# Simple and Optimal Sublinear Algorithms for Mean Estimation

**Beatrice Bertolotti**
University of Pavia, Italy
`beatrice.bertolotti02@universitadipavia.it`

**Matteo Russo**
Sapienza University of Rome, Italy
`mrusso@diag.uniroma1.it`

**Chris Schwiegelshohn**
Aarhus University, Denmark
`schwiegelshohn@cs.au.dk`

**Sudarshan Shyam**
Aarhus University, Denmark
`shyam@cs.au.dk`

## Abstract

We study the sublinear multivariate mean estimation problem in $d$-dimensional Euclidean space. Specifically, we aim to find the mean $\mu$ of a ground point set $A$, which minimizes the sum of squared Euclidean distances of the points in $A$ to $\mu$. We first show that a multiplicative $(1 + \varepsilon)$ approximation to $\mu$ can be found with probability $1 - \delta$ using $O(\varepsilon^{-1} \log \delta^{-1})$ many independent uniform random samples, and provide a matching lower bound. Furthermore, we give two estimators with optimal sample complexity that can be computed in optimal running time for extracting a suitable approximate mean:

1. The coordinate-wise median of $\log \delta^{-1}$ sample means of sample size $\varepsilon^{-1}$. As a corollary, we also show improved convergence rates for this estimator for estimating means of multivariate distributions.

2. The geometric median of $\log \delta^{-1}$ sample means of sample size $\varepsilon^{-1}$. To compute a solution efficiently, we design a novel and simple gradient descent algorithm that is significantly faster for our specific setting than all other known algorithms for computing geometric medians.

In addition, we propose an order statistics approach that is empirically competitive with these algorithms, has an optimal sample complexity and matches the running time up to lower order terms.

We finally provide an extensive experimental evaluation among several estimators which concludes that the geometric-median-of-means-based approach is typically the most competitive in practice.

## 1 Introduction

An extremely simple algorithmic paradigm is to sample a subset of the input independently and uniformly at random and solve a problem on the sample. Such algorithms are called sublinear algorithms and form a staple of large data analysis. The most desirable property of a sublinear algorithm is that the query of sample complexity is as small as possible. For most problems, it is rare for a sublinear algorithm with only constant queries to produce multiplicative guarantees. Remarkably, finding the multivariate mean of a point set $A$ in Euclidean space is one of the cases where not only can we obtain a multiplicative error with constant queries, but can do so with arbitrary precision. Specifically, we are interested in the objective

$$\min_{c \in \mathbb{R}^d} \sum_{p \in A} \|p - c\|^2,$$

39th Conference on Neural Information Processing Systems (NeurIPS 2025).

where $\|x\| = \sqrt{\sum_{i=1}^{d} x_i^2}$ denotes the Euclidean norm. It is well known that the mean of the point set $A$ minimizes this expression. We say that $c$ is a $(1 + \varepsilon)$-approximate mean, if the cost of $c$ is within a $(1 + \varepsilon)$ factor of the cost of the optimal mean.

The exact sample complexity for obtaining approximate means is still an open problem. Inaba et al. [1994] showed that $\varepsilon^{-1}$ uniform and independent samples are necessary and sufficient for the empirical mean to be a $(1 + \varepsilon)$-approximate mean on expectation. To extend this to a high probability guarantee, that is, a success probability of $1 - \delta$, $O(\varepsilon^{-1}\delta^{-1})$ samples are sufficient. Unfortunately, this bound is tight when using the empirical mean (see Appendix E). Cohen-Addad et al. [2021] improved the dependence on the probability of failure by achieving a sample complexity of $\tilde{O}(\varepsilon^{-5}\log^2 \delta^{-1})$, at the cost of a significantly worse dependency on $\varepsilon$. In the same paper, the authors also gave a lower bound of $\Omega(\varepsilon^{-1})$ for any sublinear algorithm succeeding with constant probability, showing that the result by Inaba et al. [1994] is optimal in the constant success probability regime. Subsequently, Musco et al. [2022] presented an algorithm with $O\left(\varepsilon^{-1}d(\log^2 d \cdot \log(d\varepsilon^{-1}) + \log \delta^{-1})\log \delta^{-1}\right)$ queries, which combines the tight dependency on $\varepsilon$ from Inaba et al. [1994] with the improved dependency on $\delta$ from Cohen-Addad et al. [2021], at the cost of a dependency on the dimension. Recently, Woodruff and Yasuda [2024] achieved almost best of all worlds trade-offs with a sample complexity of $O(\varepsilon^{-1}(\log \varepsilon^{-1} + \log \delta^{-1})\log \delta^{-1})$.

## 1.1  Our Contributions

Our main contributions are three algorithms that estimate the mean of a point set and that have optimal sample complexity. In addition, the running times are linear in the sample complexity for one and almost linear for the other two and thus optimal. In the first step of the algorithms, see also Algorithm 1, we draw $O(\varepsilon^{-1}\log \delta^{-1})$ points uniformly at random. We then split the sample into $O(\log \delta^{-1})$ subsamples of size $O(\varepsilon^{-1})$, compute the empirical mean for each subsample, and run an aggregation procedure, i.e., a subroutine $\textsc{Aggregate}(P)$ that combines the sample set produced to output an estimate of the mean.

---

**Algorithm 1** $\textsc{MeanEstimate}(\varepsilon, \delta)$

---

**for** $i = 1, \ldots, b\log \delta^{-1}$ **do**
    Sample $S_i$ points independently and uniformly at random with $|S_i| = a\varepsilon^{-1}$
    Compute the sample mean $\hat{\mu}_i = \frac{1}{|S_i|} \cdot \sum_{p \in S_i} p$
Output $\textsc{Aggregate}(\hat{\mu}_1, \ldots, \hat{\mu}_{b\log \delta^{-1}})$

---

The sample complexity of this algorithm, $m = b\log \delta^{-1} \cdot a\varepsilon^{-1}$ turns out to be optimal, as shown in Section E. Phrased as a generalization question for bounding the excess risk given independent samples $S$ from an underlying arbitrary but fixed distribution, we show an optimal convergence rate of $\Theta\left(\frac{\log \delta^{-1}}{|S|}\right)$.

The remaining question is to find a good solution (i.e., run $\textsc{Aggregate}$) quickly. Clearly, the sample complexity is also a lower bound for the running time of any algorithm. Achieving this running time and in particular, breaking through the $O(\log^2 \delta^{-1})$ barrier inherent in all previous algorithms, is a substantial challenge, even for computing any constant approximation. It is fairly easy to show that most of the sample means are good estimators for the true mean. Selecting one of the good estimators is nevertheless difficult, as we have no way of estimating the cost and thus no easy way of distinguishing successful estimators from unsuccessful ones. Instead, we propose three algorithm to perform such a task.

**Coordinate-Wise-Median-of-Means and Geometric-Median-of-Means.**  Both the coordinate-wise-median-of-means and the geometric median-of-means are natural generalizations of the median-of-means estimator in a single dimension. It turns out both have optimal sample complexity, up to constant factors. The former can be straightforwardly computed in linear time. Achieving a good running time for the latter is more challenging. Black box applications of gradient descent methods do not offer fast running times to compute suitable medians in our setting, but we show that a relatively simple gradient descent (see Algorithm 2) finds a suitable solution in linear time.

Using theoretical benchmarks, it is not possible to determine the more favorable estimator and it is not too difficult to construct instances where one outperforms the other. We therefore conduct an extensive experimental evaluation on real world data sets to measure the performance of these two estimators, along with previously proposed algorithms and our third contribution, the MinSumSelect algorithm.

**MinSumSelect.** Our other algorithm, given in Algorithm 3 (in Appendix B), uses a carefully chosen order statistic combined with clustering ideas. Specifically, the observation is that the sum of distances of the $t$ closest points allows us to distinguish between a successful estimate and an unsuccessful estimate, for an appropriate choice of $t$. Here, the running time improves to $O((\varepsilon^{-1} + \log^\gamma \delta^{-1}) \cdot d \log \delta^{-1})$, for any constant $\gamma > 0$. The algorithm can also be parameterized to achieve a similar running time to the gradient descent algorithm, at the cost of increasing the sample size by poly $\left(\log \log \delta^{-1}\right)$ factors.

## 1.2  Related Work

We covered the related work specific to sublinear algorithms for estimating the mean earlier, but there are several other closely related topics. Given a distribution, a classical question is to design good estimators for some underlying statistic. For high-dimensional means, there has been substantial attention recently, see Lee and Valiant [2022], Lugosi and Mendelson [2019, 2024] and references therein. There are common notions with our work, as the distance to the true mean and tail bounds for the estimators are an important quantity in our setting as well. Nevertheless, in this line of work, the running time, which is a very important focus for sublinear algorithms, is not a primary concern.

Another interesting line of work focuses on data dependent mean estimation, where the bounds derived are dependent on the quality of the data [Dang et al., 2023]. There are limited regimes in which estimators with better than sub-Gaussian estimators are possible. This is especially interesting given the recent efforts in the direction of beyond worst-case analysis of algorithms.

In a more general setting, one could ask for approximate power means minimizing the objective $\min_{c \in \mathbb{R}^d} \sum_{p \in A} \|p - c\|^z$ for some positive integer $z$. For $z > 2$, this problem was first posed by Cohen-Addad et al. [2021] who showed that $\tilde{O}(\varepsilon^{-z-3} \log^2 \delta^{-1})$ samples were sufficient for a high success probability while $\Omega(\varepsilon^{-z+1})$ samples were necessary for constant success probability. Woodruff and Yasuda [2024] improved this even further, obtaining a sample complexity of $O(\varepsilon^{-z+1}(\log \varepsilon^{-1} + \log \delta^{-1}) \log \delta^{-1})$. Perhaps surprisingly, the case $z = 1$, also known as the geometric median requires $\Theta(\varepsilon^{-2} \log \delta^{-1})$ many samples, see Chen and Derezinski [2021], Parulekar et al. [2021] for lower bounds and Cohen et al. [2016] for an optimal algorithm. Finally, the special case $z \to \infty$ corresponds to the minimum enclosing ball problem. The solution is highly susceptible to outliers, thus any sublinear algorithm must either be allowed to discard a fraction of the input, or rely on additive guarantees (see Clarkson et al. [2012] and Ding [2020] for examples).

The sample complexity for the $k$-means problem has been extensively studied in the context of generalization bounds. Here, we are given an arbitrary but fixed distribution $\mathcal{D}$ typically supported on the unit Euclidean sphere, we aim to find a set of $k$ centers $C$ minimizing $\int_p \min_{c \in C} \|p - c\|^2 \mathbb{P}_\mathcal{D}[p] dp$. Following a long line of work, the best currently known learning rates with a sample size $n$ for this problem are of the order $O\left(\sqrt{\frac{k \log k}{|S|}} + \frac{\log \delta^{-1}}{|S|}\right)$, see Bartlett et al. [1998], Clemençon [2011], Cohen-Addad et al. [2025], Fefferman et al. [2016], Klochkov et al. [2021], Linder [2000], Narayanan and Mitter [2010], Pollard and references therein. Interestingly, the best known lower bounds on the learning rates for arbitrary values of $k$ are at least $\Omega\left(\sqrt{\frac{k}{|S|}}\right)$, see Bartlett et al. [1998], Bucarelli et al. [2023].

Another related line of work is on coresets. For center-based objectives, a coreset approximates the cost of *any* given center with respect to the sum of distances raised to some power, where $z = 2$ is the power related to the 1-means objective. While there are some exceptions that obtain very small coreset sizes via other techniques, see Afshani and Schwiegelshohn [2024], Braverman et al. [2022], Huang et al. [2023], Maalouf et al. [2021, 2022], the sample complexity of sensitivity sampling has by far been the most widely studied complexity measure in this line of work, see Bansal et al. [2024], Braverman et al. [2021], Cohen-Addad et al. [2021], Feldman and Langberg [2011], Langberg and Schulman [2010].

**Further Comments and Limitations.** We assume that the samples are drawn independently from a data set, and that there are no corruptions. Efficiency and scalability are not limitations, as the present work only has optimal results in that regard. Regarding notions of privacy: Private mean estimation is an important and well studied problem, but if we wish a purely multiplicative approximation, as we aim to do here, no privacy guarantees are possible. Regarding fairness notions, sublinear algorithms cannot be individually fair in a meaningful way, as we never access the entire data set. Other clustering fairness notions, such as representational fairness, do not apply for a single cluster.

## 2   Preliminaries

For a set of points $A \subset \mathbb{R}^d$, whose cardinality is $|A| = n$, we use $\mu(A) := \frac{1}{n} \cdot \sum_{p \in A} p$ to denote the mean, using $\mu$ if $A$ is clear from the context. Denote by $\mathrm{OPT} = \min_{c \in \mathbb{R}^d} \sum_{p \in A} \|p - c\|^2$. In addition, for a set of points $S$ sampled uniformly at random from $A$, we call $\hat{\mu}(S)$ the empirical estimator of $\mu(A)$, or simply $\hat{\mu}$ if $S$ is clear from context. A useful relationship between the goodness of $\hat{\mu}$ and the size of $|S|$ is established via the following identity.

**Lemma 2.1** (Lemma 1 of Inaba et al. [1994]). $\mathbb{E}\left[\|\mu(A) - \hat{\mu}(S)\|^2\right] = \frac{1}{|S|} \cdot \frac{\mathrm{OPT}}{n}$.

At the heart of most algorithms for Euclidean means lies the following (arguably folklore) identity.

**Lemma 2.2** (High Dimensional Mean-Variance Decomposition). *For any set of points $A \subset \mathbb{R}^d$ and any $c \in \mathbb{R}^d$, we have $\sum_{p \in A} \|p - c\|^2 = \sum_{p \in A} \|p - \mu\|^2 + n \cdot \|\mu - c\|^2$.*

Lemma 2.2 implies that $\hat{\mu}$ is a $(1 + \varepsilon)$ appoximation if and only if $\|\hat{\mu} - \mu\| \leq \sqrt{\frac{\varepsilon \mathrm{OPT}}{n}}$, which we will often use throughout this paper. For space reasons, some proofs are omitted in the main body and deferred to the appendix.

For the proofs in the next sections, we require a basic probabilistic lemma. We say that an empirical mean $\hat{\mu}_i$ is *good*, if $\|\hat{\mu}_i - \mu\| \leq r$, where $r = \frac{1}{11}\sqrt{\frac{\varepsilon \mathrm{OPT}}{n}}$. We denote the set of *good* sample means by $G$, and define event

$$\mathcal{E} := \left\{ |G| \geq \frac{7}{10} \cdot b \log \delta^{-1} \right\}. \tag{1}$$

The following lemma is a straightforward application of the Chernoff bound.

**Lemma 2.3.** *With probability $1 - \delta$, event $\mathcal{E}$ holds, for a sufficiently large absolute constants $a$ and $b$.*

*Proof.* We set the values of the constants to be $a = 1440$, $b = 50$ and $r = \frac{1}{11}\sqrt{\frac{\varepsilon \mathrm{OPT}}{n}}$ (for reasons which will be clear in the proof). First, we show that a sample mean of $a\varepsilon^{-1}$ samples is good with probability at least $0.9$. This follows from Lemma 2.1 and Markov's inequality.

$$\mathbb{P}\left[ \|\hat{\mu}_i - \mu\|^2 \leq r^2 \left( -\frac{\varepsilon \mathrm{OPT}}{121 n} \right) \right] \geq 0.9.$$

Applying the standard Chernoff bound (with $b = 50$), we get

$$\mathbb{P}\left[ |G| \leq \frac{7}{10} \cdot b \log \delta^{-1} \right] \leq \exp\left( -\frac{2}{81} \cdot b \log \delta^{-1} \right) \leq \delta,$$

where the $b$ is chosen so that $-\frac{2b}{81} < -1$. $\qquad \square$

## 3   Coordinate-Wise Median-of-Means

In this section, we show that the coordinate-wise median-of-means is an optimal estimator and thus leads to a linear time algorithm for mean estimation. That is, at the end of Algorithm 1 we simply

use AGGREGATE$(\hat{\mu}_1, \ldots, \hat{\mu}_{b \log \delta^{-1}})$ = COORDWISEMEDIAN$(\hat{\mu}_1, \ldots, \hat{\mu}_{b \log \delta^{-1}})$ as an aggregation procedure. We first do so in the context of sublinear algorithms and then extend the result to the case of distributional mean estimation in high dimensions.

### 3.1 Coordinate-Wise Median-of-Means in the Sublinear Model

We prove the following theorem:

**Theorem 3.1.** *Algorithm 1 run with $\nu_{\text{CWM}} :=$ COORDWISEMEDIAN$(\hat{\mu}_1, \ldots, \hat{\mu}_{b \log \delta^{-1}})$ as an aggregation routine outputs a $(1 + \varepsilon)$-approximate Euclidean mean with probability $1 - \delta$. The running time is $O\left(\varepsilon^{-1} \cdot d \log \delta^{-1}\right)$.*

*Proof.* Consider $\nu_{\text{CWM}}$, the coordinate-wise median of all the sample means. For each coordinate $k$, let $L_k$ and $R_k$ be the sets of sample means such that $\hat{\mu}_{i,k} \leq \nu_{\text{CWM},k}$ and $\hat{\mu}_{i,k} \geq \nu_{\text{CWM},k}$ respectively. We know that $L_k, R_k$ have cardinality at least $\frac{b \log \delta^{-1}}{2}$. Depending on whether $\nu_{\text{CWM},k} > \mu_k$ or not, we show that at least one of the following statements is true:

1. For all $\hat{\mu}$ in $L_k$, we have that $|\hat{\mu}_{i,k} - \mu_k| \geq |\nu_{\text{CWM},k} - \mu_k|$.

2. For all $\hat{\mu}$ in $R_k$, we have that $|\hat{\mu}_{i,k} - \mu_k| \geq |\nu_{\text{CWM},k} - \mu_k|$.

We have $|L_k \cup R_k| = b \log \delta^{-1}$ which is the total number of sample means. First, from the definitions of $L_k$ and $R_k$, we observe that there are points (such that $\hat{\mu}_k = \nu_{\text{CWM},k}$) which belong to both. Along with the definition of the coordinate-wise median, we have $|L_k|, |R_k| \geq b \log \delta^{-1}/2$. Note that if there are no points such that $\hat{\mu}_k = \nu_{\text{CWM},k}$, then $|L_k| = |R_k| = b \log \delta^{-1}/2$.

We provide a case analysis as to why at least $1/5$ fraction of the samples are good and satisfy $|\hat{\mu}_{i,k} - \mu_k| \geq |\nu_{\text{CWM},k} - \mu_k|$.

First, we note that when the good event $\mathcal{E}$ holds, at least $1/5$ fraction of the means are good in both the sets $L_k$ and $R_k$. This follows from the fact that at least $7/10$ fraction of the means are good and the number of good means in $L_k/R_k$ is at least $7/10 - 1/2 = 1/5$.

**Case I:** $\mu_k \leq \nu_{\text{CWM},k}$. In this case, we have for all means in $R_k$, $|\hat{\mu}_{i,k} - \mu_k| \geq |\nu_{\text{CWM},k} - \mu_k|$.

**Case II:** $\mu_k > \nu_{\text{CWM},k}$. In this case, we have for all means in $L_k$, $|\hat{\mu}_{i,k} - \mu_k| \geq |\nu_{\text{CWM},k} - \mu_k|$.

From the above statements, we have for at least one of $L_k$ and $R_k$, we have that at least $1/5$ fraction of the means are good and satisfy $|\hat{\mu}_{i,k} - \mu_k| \geq |\nu_{\text{CWM},k} - \mu_k|$.

From Lemma 1, we know that at least $\frac{7}{10}$ fraction of the means are *good* (i.e., $\|\hat{\mu}_i - \mu\| \leq r$) with probability at least $1 - \delta$. Hence, we infer that at least $\frac{7}{10} - \frac{1}{2} = \frac{1}{5}$ of the sample means are *good* and satisfy $|\hat{\mu}_{i,k} - \mu_k| \geq |\nu_{\text{CWM},k} - \mu_k|$. On average,

$$\frac{1}{|G|} \sum_{i \in G} |\hat{\mu}_{i,k} - \mu_k|^2 \geq \frac{|\nu_{\text{CWM},k} - \mu_k|^2}{5}.$$

Summing over all coordinates $k$ gives

$$\sum_{k \in [d]} \frac{1}{|G|} \sum_{i \in G} |\hat{\mu}_{i,k} - \mu_k|^2 \geq \sum_{k \in [d]} \frac{|\nu_{\text{CWM},k} - \mu_k|^2}{5} = \frac{\|\nu_{\text{CWM}} - \mu\|^2}{5}.$$

Interchanging the sums on the left-hand side, we get

$$\|\nu_{\text{CWM}} - \mu\|^2 \leq \frac{5}{|G|} \sum_{i \in G} \sum_{k \in [d]} |\hat{\mu}_{i,k} - \mu_k|^2 = \frac{5}{|G|} \sum_{i \in G} \|\hat{\mu}_i - \mu\|^2 \leq 5r^2.$$

The theorem follows from the definition of $r = \frac{1}{11} \sqrt{\frac{\varepsilon \text{OPT}}{n}}$, and the running time follows by recognizing that we take $O(\log \delta^{-1})$ sample means, each composed of $O(\varepsilon^{-1})$ many points in $d$ dimensions. $\qquad \square$

## 3.2 Learning the Mean of a High-Dimensional Distribution

In this section, we relate the results in the literature with respect to mean estimation of high dimensional multivariate probability distributions and place our results in the context of these works.

We first describe the setting. The goal is to estimate the mean $\mu$ of a probability distribution $\mathcal{D}$ in $\mathbb{R}^d$. A standard assumption is the existence of a covariance matrix $\Sigma$. We want the estimate to be close to the the true mean $\mu$ and the Euclidean distance to the mean is used as the objective. Given $N$ i.i.d. samples $X_1, X_2, \ldots, X_N \sim \mathcal{D}$, the goal is to construct an estimator $\nu = \nu(X_1, X_2, \ldots, X_N)$ such that, for any confidence interval $\delta \in [0, 1]$, with probability at least $1 - \delta$, we have $\|\mu - \nu\| \leq \varepsilon(N, \delta, \Sigma)$. The function $\varepsilon(N, \delta, \Sigma)$ is referred to as the rate of convergence of error henceforth.

The coordinate-wise median-of-means, while not optimal, is a simple and natural extension of the algorithm for the 1-dimension. A simple analysis, found in several works, see Lugosi and Mendelson [2019] and Minsker [2015], applies a union bound over all dimensions and concludes that if the mean is concentrated along every dimension, it will be concentrated in general. This analysis proves a convergence rate of the order $O\left(\sqrt{\frac{\text{TR}(\Sigma) \log(d\delta^{-1})}{N}}\right)$, where $\text{TR}(\Sigma)$ is the trace of the covariance matrix. Somewhat misleadingly, the dependency on $\log d$ is often implied to be necessary.

The analysis of the coordinate-wise median-of-means for sublinear algorithms can be adapted to yield convergence rate for the mean estimation problem. Specifically, the coordinate-wise median-of-means estimator has the following, dimension-free convergence rate.

**Theorem 3.2.** *Let $X_1, X_2, \ldots, X_N$ be independent samples from distribution $\mathcal{D}$ with variance $\mu$ and covariance matrix $\Sigma$, then the coordinate-wise median-of-means estimator $\nu_{\text{CWM}}$, with probability at least $1 - \delta$, satisfies*

$$\|\nu_{\text{CWM}} - \mu\| \leq 40\sqrt{\frac{\text{TR}(\Sigma) \log \delta^{-1}}{N}}$$

The proof is an adaptation of the proof for our setting and the details are deferred to the appendix.

## 4 Geometric Median-of-Means and Gradient Descent

In the previous section, we showed that the coordinate-wise median of the sample means is a $(1 + \varepsilon)$-approximation of the true mean and can be computed in linear time in the number of samples. However, one could generate instances where the coordinate-wise median is consistently a worse approximation to the true mean than the geometric median is, and vice-versa. This is shown in Appendix F. Therefore, we prove that the geometric median-of-means estimator also has an optimal sample complexity and give a gradient descent algorithm (Algorithm 2) for computing a sufficiently good estimate of it efficiently. It is important to note that Algorithm 2 does not always converge to the geometric median of the points. Take, for example, the triangle with coordinates $(0, 1), (0, -1), (1, 0)$. The geometric median is the point $(\frac{1}{\sqrt{3}}, 0)$. If we initialize the algorithm at the origin, the algorithm never makes any progress and stays at the origin.

The gradient of the geometric median objective $\sum_{p \in P} \|p - c\|$ at any point $q$ with respect to some reference point set $P$ is $\nabla(q) = \sum_{p \in P} \nabla_p(q)$, where $\nabla_p(q) := \frac{q-p}{\|q-p\|}$ is the contribution to the gradient from point $p$ (excluding co-located points $q$ to make it well defined).

---

**Algorithm 2** FASTGD$(P)$

---

Compute $\nu_{\text{CWM}} = \text{COORDWISEMEDIAN}(\hat{\mu}_1, \ldots, \hat{\mu}_{b \log \delta^{-1}})$
**for** $j = 1, \ldots, T$ **do**
    Compute the gradient $\nabla(c_{j-1}) = \sum_{p \in P} \frac{c_{j-1}-p}{\|c_{j-1}-p\|}$
    Project all $p_i \in P$ onto $c_{j-1} - \nabla(c_{j-1})$, denoting the projection by $p_{i,c_{j-1}}$
    Compute the 1-dimensional median $c_j$ of the $p_{i,c_{j-1}}$'s along the line $c_{j-1} - \nabla(c_{j-1})$
Output $c_T$

---

The proof that the geometric median-of-means estimator is optimal is algorithmic, i.e. it will follow as a result of the analysis of Algorithm 2. We initialize Algorithm 2 with $P = \{\hat{\mu}_1, \ldots, \hat{\mu}_{b \log \delta^{-1}}\}$ and

the coordinate-wise median $\nu_{\text{CWM}}$ of $P$ as the starting estimate. The goal is to prove the following theorem:

**Theorem 4.1.** *Algorithm 1 run with Algorithm 2 as an aggregation routine outputs a $(1 + \varepsilon)$-approximate Euclidean mean with probability $1 - \delta$ if $T \geq \gamma$ for some absolute constant $\gamma$. The running time is $O\left(\varepsilon^{-1} \cdot d \log \delta^{-1}\right)$.*

The proof of Theorem 4.1 is articulated in two main steps. First, we show that, given a candidate estimate $c_j$ is very far from $\mu$, the gradient of the geometric median of means evaluated at $c_j$ points in direction of $\mu$. Second, we must determine a good updated solution along this gradient. Line-sweeps or second-order methods are all possible candidates, but computationally expensive. Instead, we perform and update by selecting the median-of-means along the gradient. Third, we show that once the algorithm determines a solution close to $\mu$, the gradient updates will remain close thereafter.

**Step 1: Fast convergence to close estimates from far ones.** In order to estimate the mean quickly with gradient descent, we will make use of the following geometric median gradient properties whenever our current estimate $c_j$ is far from $\mu$.

**Lemma 4.2.** *Conditioned on event $\mathcal{E}$, we have:*

(a) *For any point $q \in \mathbb{R}^d$ with $\|q - \mu\| > 10r$, then $\|\nabla(q)\| > \frac{3}{10} \cdot b \log \delta^{-1}$;*

(b) *If, at iteration $j$, Algorithm 2 chooses a point $c_j$ such that $\|c_j - \mu\| \geq 10r$, then $\|c_{j+1} - \mu\| \leq \frac{7}{10} \cdot \|c_j - \mu\|$.*

*Proof.* We first show that every vector corresponding to a *good* mean is approximately in the same direction from $q$. Let $\hat{\mu}$ be a good sample mean. Consider the triangle $\{q, \mu, \hat{\mu}\}$ and let $\alpha_i$ and $\beta_i$ be the angle in $q$ and $\hat{\mu}$ respectively. We show an upper-bound for $\cos(\alpha_i)$. Using the law of sines,

$$\frac{\sin(\alpha_i)}{\|\mu - \hat{\mu}_i\|} = \frac{\sin(\beta_i)}{\|q - \mu\|} \leq \frac{1}{\|q - \mu\|}$$

We get $\sin(\alpha_i) \leq \frac{\|\mu - \hat{\mu}_i\|}{\|q - \mu\|} \leq \frac{1}{10}$, which implies $\cos(\alpha_i) \geq \sqrt{1 - \sin^2(\alpha_i)} \geq 0.99$. To bound $\|\nabla(q)\|$, we consider the contribution of *good* and *bad* means separately. We write $\nabla(q) = \sum_{i \in [b \log \delta^{-1}]} \nabla_i(q) = \nabla_G(q) + \nabla_B(q)$, where $\nabla_G, \nabla_B$ respectively denote good and bad points contributions to the gradient evaluated at $q$ (see Figure 1).

$$\|\nabla_G(q)\| = \left\| \sum_{i:\ \hat{\mu}_i \text{ good}} \nabla_i(q) \right\| \geq \left\| \sum_{i:\ \hat{\mu}_i \text{ good}} \cos(\alpha_i) \cdot \frac{\mu - q}{\|\mu - q\|} \right\| \geq \frac{7}{10} \cdot 0.99 \cdot b \log \delta^{-1},$$

where the first inequality holds because projecting the gradient onto any direction only decreases the norm. Conversely, the contribution given by bad sample means to the gradient norm is at most

$$\|\nabla_B(q)\| = \left\| \sum_{i:\ \hat{\mu}_i \text{ not good}} \nabla_i(q) \right\| \leq \frac{3}{10} \cdot b \log \delta^{-1}$$

where the inequality follows from the upper-bound on the number of *bad* means when $\mathcal{E}$ holds and the triangle inequality. Part (a) follows by combining both bounds: We have, due to the triangle inequality, an overall lower bound on the norm of the gradient of

$$\left\| \sum_i \nabla_i(q) \right\| \geq \left\| \sum_{i:\ \hat{\mu}_i \text{ good}} \nabla_i(q) \right\| - \left\| \sum_{i:\ \hat{\mu}_i \text{ not good}} \nabla_i(q) \right\| \geq \frac{3}{10} \cdot b \log \delta^{-1}.$$

For part (b), similarly to the above, we focus on the triangle $\{q, q - \nabla_G, q - \nabla_G - \nabla_B\}$. Let $\beta$ be the angle in vertex $q$ (see Figure 1, with $q$ instead of $c_j$) and let $\theta$ be the angle in vertex $q - \nabla_G - \nabla_B$. Using the bounds of $\|\nabla_G\|$ and $\|\nabla_B\|$ and the law of sines, we get

$$\frac{\sin(\beta)}{\|\nabla_B\|} = \frac{\sin(\theta)}{\|\nabla_G\|} \leq \frac{1}{\|\nabla_G\|},$$

which implies that $\sin(\beta) \leq \frac{\|\nabla_B\|}{\|\nabla_G\|} \leq \frac{1}{2}$ .

Next, we upper-bound $\|c_{j+1} - \mu\|$ to prove part (b). Consider the projection of $\mu$ onto $c_j - \nabla_G(c_j)$ (the green line in Figure 1), call it $y$. Now, let us consider the projection of $y$ onto $c_j - \nabla(c_j)$, call it $\text{proj}_j(y)$. Since the good samples are in the majority, the median $c_{j+1}$ of the projected sample means (including both good and bad) lies within the convex hull of the good projected means. This holds because, under any projection, the median must be between two good points (analogously to the one-dimensional median-of-means estimator). This implies that, for $c_{j+1}$, we have

$$\|c_{j+1} - \mu\| \leq \|c_{j+1} - y\| + \|y - \mu\| \leq \underbrace{\|c_{j+1} - \text{proj}_j(y)\|}_{\leq r} + \underbrace{\|\text{proj}_j(y) - y\|}_{=\sin(\beta) \cdot \|c_j - y\|} + \underbrace{\|y - \mu\|}_{\leq r}$$

$$\leq \sin(\beta) \cdot \|c_j - \mu\| + 2r \leq \frac{7}{10} \cdot \|c_j - \mu\|,$$

where the third inequality follows from the fact that $\|c_j - y\| \leq \|c_j - \mu\|$ (given that $c_j - \mu$ and $c_j - y$ are respectively the hypotenuse and the leg of the right-angle triangle $\{c_j, \mu, y\}$) and $\|c_j - \mu\| \geq 10r$. □

We wish to briefly remark on the sample complexity of the geometric median-of-means estimator and generalization bounds. Optimality follows from part (a) of Lemma 4.2, as if the gradient of the geometric median $\nu_{\text{GM}}$ is 0, $\|\nu_{\text{GM}} - \mu\| \leq 10r$ must hold.

**Corollary 4.3.** *For sufficiently large absolute constants $a$ and $b$, the geometric median $\nu_{\text{GM}} := \underset{c \in \mathbb{R}^d}{\arg\min} \sum_{i=1}^{b \log \delta^{-1}} \|\hat{\mu}_i - c\|$ of $b \log \delta^{-1}$ many sample means consisting of $a\varepsilon^{-1}$ many points is a $(1 + \varepsilon)$-approximate mean with probability at least $1 - \delta$.*

**Remark 4.4.** *The geometric median cannot be computed exactly, we can merely approximate it. The quality of the choice of approximation is important here. Using event $\mathcal{E}$ by itself, it is not clear that even a 2-approximate geometric median will recover a $(1 + O(\varepsilon))$-approximate mean. Indeed, if we assume that a single $\hat{\mu}_i$ has distance $\sqrt{\frac{\varepsilon \text{OPT}}{\delta n}}$ to the optimal mean, we must compute a $(1 + \delta^{-O(1)})$-approximate median. In Appendix E, we will show that this assumption is warranted for worst case instances. The fastest algorithm for finding a $(1 + \gamma)$-approximate geometric median by Cohen et al. [2016] runs in time $O(nd \log^3 \gamma^{-1})$. In our setting, $n = b \log \delta^{-1}$. The resulting running time of $O(\log^4 \delta^{-1} \cdot d)$ is thus substantially slower that the other three algorithms presented in this paper.*

**Step 2: Once the estimate is close, it remains close.**

**Lemma 4.5.** *Conditioned on event $\mathcal{E}$, if, at iteration $j$, Algorithm 2 chooses a point $c_j$ such that $\|c_j - \mu\| \leq 10r$, then $\|c_{j+1} - \mu\| \leq 11r$.*

*Proof.* Consider a point $c_j$ chosen by Algorithm 2 such that $\|c_j - \mu\| \leq 10r$. Its gradient vector is $\nabla(c_j)$ and Algorithm 2 projects all other sample means onto $c_j - \nabla(c_j)$ and takes the median to be $c_{j+1}$. For every mean $\hat{\mu}$, we denote its projection by $\text{proj}(\hat{\mu})$. We make two observations. First, the set of $\text{proj}(\hat{\mu})$ of all *good* points lie on a bounded line segment of length at most $2r$ (see Figure 1). Second, due to the event $\mathcal{E}$, we have that the median chosen lies within the bounded line segment.

From the second observation, we have $\|c_{j+1} - \text{proj}(\mu)\| \leq r$, which is the radius of the *good* ball (see Figure 1). Applying triangle inequality, we get

$$\|c_{j+1} - \mu\| \leq \|c_{j+1} - \text{proj}(\mu)\| + \|\text{proj}(\mu) - \mu\| \leq r + \|c_j - \mu\| \leq 11r,$$

where the second inequality follows from $\text{proj}(\mu)$ being a projection on a line passing through $c_j$. □

*Proof of Theorem 4.1.* We condition on event $\mathcal{E}$. In Algorithm 2, the initial value $\nu_{\text{CWM}}$ is taken to be the coordinate-wise median of all the $b \log \delta^{-1}$ sample means. From Theorem 3.1, we know that $\|\nu_{\text{CWM}} - \mu\| \leq \sqrt{5}r$. Using part (b) of Lemma 4.2, we have that as long as $\|c_j - \mu\| \geq 10r$, it holds that $\|c_{j+1} - \mu\| \leq \frac{7}{10} \cdot \|c_j - \mu\|$. As the quantity $\|c_j - \mu\|$ is decreasing exponentially, it follows that for some absolute constant $\gamma$ there exists an iteration $j \leq \gamma$, such that $\|c_j - \mu\| \leq 10r$. In the next

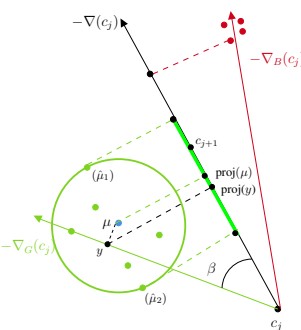

Figure 1: Good empirical means are represented in green and bad ones in red. The ball of good means centered at $\mu$ has radius $r$. Projection of all the good means lie on the bounded line segment of length at most $2r$.

iterations of the algorithm, from Lemma 4.5, we know that for the terminating value of the algorithm, $c_T$ satisfies $\|c_T - \mu\| \le 11r$, which yields the desired approximation ratio.

Concerning the the running time, we first need to compute $b \log \delta^{-1}$ sample means from $a\varepsilon^{-1}$ many uniform samples, and then, in each of the $\gamma$ iterations, there are $b \log \delta^{-1}$ sample means to project onto the gradient vector, an operation that takes $O(d \log \delta^{-1})$ time, with $d$ being the dimension of the ambient space. To compute the median of all projections in every iteration, as well as finding the initial coordinate-wise median $\nu_{\mathrm{CWM}}$ can be done using a linear-time rank select procedure (see [Cormen et al., 2009, Chapter 9.3]). For each iteration, as well as the initialization, this therefore takes $O(d \log \delta^{-1})$ time. Thus, the overall running time is $O\left(\varepsilon^{-1} \cdot d \log \delta^{-1}\right)$. $\qquad\square$

## 5 Experimental Evaluation

We conclude this paper with a short experimental evaluation. Code base and results can be found at `https://github.com/matteorusso/sublinear_mean_estimation`. The experiments were carried out on the Google Colab default CPU. Our goal is to understand the practical viability of various estimators, such as the three optimal estimators included here, as well as the previous results from Cohen-Addad et al. [2021], Woodruff and Yasuda [2024] and the empirical mean of the samples. In particular, we wish to understand if the gradient updates from Algorithm 2 improve over an initialization given by the coordinate-wise median. It is not difficult to generate instances where one of these algorithms outperforms the other, but ultimately the empirically best estimator can only be determined via experiments on real-world data.

**Algorithms.** The reference algorithms for fast sublinear mean estimation is the 1-means coreset-based CSS algorithm Cohen-Addad et al. [2021], the active-regression coreset-based WY algorithm Woodruff and Yasuda [2024], and our Algorithms 2 and 3. In addition, we also used the standard empirical mean as a baseline. Each of these algorithms are given a set of $m$ points sampled uniformly at random from the underlying point set and compared with respect to running time and quality of the solution.

We implement both MINSUMSELECT and FASTGD. As described in Section 4, we take as initial guess the coordinate-wise median of the computed sample means. We also simply compare against the coordinate-wise median initialization step of FASTGD, referred to as COORDWISEMEDIAN.

**Data Sets, Setup and Results.** We test our algorithms against the benchmarks mentioned above on the following datasets: MNIST and Fashion-MNIST, both of which are composed of 60,000 points, each with 784 features. We also consider CoverType, composed of 581,012 points, each with 54 features. We describe the gist of the results here, more details such as numerical figures are given in Appendix G. For each sample size $m \in \{10, 15, 20, 25, 30, 100, 200, 500, 1000, 2000, 5000, 10000\}$, we repeat the execution of every algorithm 50 times and report averages and variances across runs. We plot the accuracy (i.e., $^{\mathrm{ALG}}/_{\mathrm{OPT}}$), and runing time as a function of the sample size for the various algorithms. Due to space constraints, we only report results on MNIST in this section (see Figure 2). The behaviour of accuracy and runtime as a function of sample size is similar across all datasets.

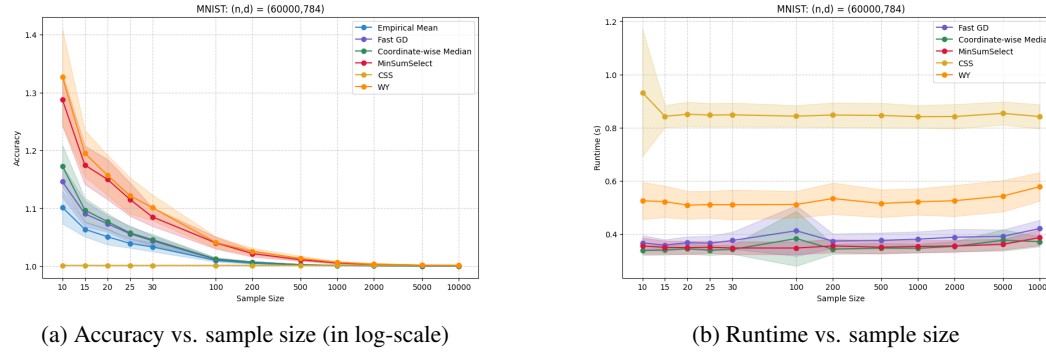

(a) Accuracy vs. sample size (in log-scale)

(b) Runtime vs. sample size

Figure 2: MNIST Dataset: Accuracy and runtime against sample size.

Our observations from the experimental evaluation on real-world datasets provide a few insights on the proposed methods. Among all algorithms, MINSUMSELECT and COORDWISEMEDIAN emerge as the fastest, consistently achieving superior runtime performance across all datasets. Their efficiency highlights their suitability for large-scale applications where computational speed is critical. In terms of accuracy, both MINSUMSELECT, COORDWISEMEDIAN and FASTGD perform comparably, with FASTGD showing a slight edge in most cases. Notably, FASTGD almost never underperforms relative to COORDWISEMEDIAN, despite both offering the same theoretical guarantees. This suggests that, in practice, applying a few gradient-based refinement steps toward the geometric median, starting from the coordinate-wise median, can yield measurable improvements.

Interestingly, despite having the weakest theoretical guarantees, the empirical mean consistently ranks among the best algorithms in terms of accuracy. Since it can also be computed extremely quickly via the closed form, it is also by far the fastest among the candidate algorithms. While there exist examples where the empirical mean does not provide a good and robust estimation, these examples do not seem to appear on any of the data sets we tried and may not be common in practice, see the appendix for a more detailed discussion. Nevertheless, the empirical mean is not consistently the best estimator, which goes to the CSS-estimator, which also requires extensive filtering and preprocessing and is among the slowest available methods. The WY algorithm performs worse in terms of accuracy. While it is not known whether the stated sample complexity upper bounds for CSS and WY are tight, these observation may indicate that these algorithms are not tightly analyzed. Additionally, they are considerably slower than our algorithms, regardless of sample size, due to inherent overhead in pre- and postprocessing steps. However, optimization is very fast for these algorithms because they simply output the mean.

In summary, the algorithms proposed in this paper are always competitive in terms of accuracy with the other methods, at least when given a moderately large sample size, while being among the fastest methods available. While the refinement via FASTGD can offer improvements in practice, the best candidate algorithms seem to be either COORDWISEMEDIAN or taking the empirical mean.

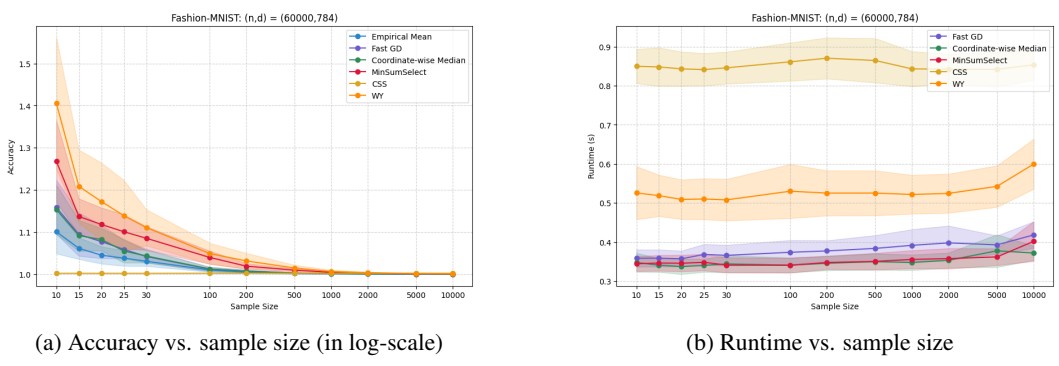

(a) Accuracy vs. sample size (in log-scale)

(b) Runtime vs. sample size

Figure 3: Fashion-MNIST Dataset: Accuracy and runtime against sample size.

## Acknowledgements

We thank the anonymous reviewer from a previous version of this paper for pointing an improvement of our analysis of the coordinate-wise median-of-means estimator. Matteo Russo was partially supported by the FAIR (Future Artificial Intelligence Research) project PE0000013, the NextGenerationEU program within the PNRR-PE-AI scheme (M4C2, investment 1.3, line on Artificial Intelligence), the PNRR MUR project IR0000013-SoBigData.it, and by the MUR PRIN grant 2022EKNE5K (Learning in Markets and Society). Chris Schwiegelshohn was supported by a Google Research Award and by the Independent Research Fund Denmark (DFF) under a Sapere Aude Research Leader grant No 1051-00106B. Sudarshan Shyam was partially supported by the Independent Research Fund Denmark (DFF) under grant 2032-00185B.

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

## A  Preliminaries

**Lemma A.1** (High Dimensional Mean-Variance Decomposition). *For any set of points $A \subset \mathbb{R}^d$ and any $c \in \mathbb{R}^d$, we have $\sum_{p \in A} \|p - c\|^2 = \sum_{p \in A} \|p - \mu\|^2 + n \cdot \|\mu - c\|^2$.*

*Proof.* We have

$$
\begin{aligned}
\sum_{p \in A} \|p - c\|^2 &= \sum_{p \in A} \|p - \mu + \mu - c\|^2 \\
&= \sum_{p \in A} \left( \|p - \mu\|^2 + \|\mu - c\|^2 - 2(p - \mu)^\top (\mu - c) \right) \\
&= \sum_{p \in A} \|p - \mu\|^2 + n \cdot \|\mu - c\|^2,
\end{aligned}
$$

where the last equality follows from $\sum_{p \in A}(p - \mu) = 0$. $\qquad\square$

## B  Mean Estimation via Order Statistics

In this section, we leverage order statistics of the candidate means so as to allow for a quicker aggregation of a suitable candidate mean.

---

**Algorithm 3** MINSUMSELECT$(P, i)$

---

**Input:** Set of points $p_1, \ldots p_{|P|}$, recursion depth $i$
**if** $i = 0$ **or** $|P| = 1$ **then**
    $W \leftarrow P$
**else**
    Split $P$ arbitrarily into $\sqrt{|P|}$-sized clusters $\{P_1, \ldots, P_{\sqrt{|P|}}\}$
    $W \leftarrow \emptyset$
    **for** each $P_j$ **do**
        $W \leftarrow W \cup \{\text{MINSUMSELECT}(P_j, i - 1)\}$
Output COMPUTEWINNER$(W)$

---

**Algorithm 4** COMPUTEWINNER$(P)$

---

**for** each $p_j \in P$ **do**
    Let $p'_j \in P$ be the $\frac{7}{10}|P|$-closest point to $p_j$
    Compute $D_j := \sum_{p \in P, \|p - p_j\| \le \|p'_j - p_j\|} \|p - p_j\|$
Output $\arg\min_{p_j \in P} D_j$

---

The main result is the following:

**Theorem B.1.** *Algorithm 1 run with Algorithm 3 as an aggregation routine outputs a $(1 + \varepsilon)$ approximation with probability $1 - \delta$, using a sample of size $O\left(100^i \varepsilon^{-1} \log \delta^{-1}\right)$, and running in time*

$$
O\left( \left( 100^i \varepsilon^{-1} + \log^{2^{-i}} \delta^{-1} \right) d \log \delta^{-1} \right),
$$

*for any non-negative integer $i$.*

The key observation is that a good mean can always be identified via COMPUTEWINNER as an estimate $\hat{\mu}_j$ minimizing the sum of distances of the $\frac{7}{10}$th means closest to $\hat{\mu}_j$. In a nutshell, any mean minimizing such a sum must be close to sufficiently many successful estimates. Unfortunately, a naive implementation of this idea takes time $O(\log^2 \delta^{-1} \cdot d)$, as proven in the following lemma.

**Lemma B.2.** COMPUTEWINNER$(P)$ *takes time $O(|P|^2 \cdot d)$.*

*Proof.* We compute all pairwise distances between the points in $P$, which takes time $O(|P|^2 \cdot d)$. To compute $D_j$, we first have to find the $\frac{7}{10}|P|$-closest point to $p_j$, which takes time $O(|P|)$ with a sufficiently good rank select procedure, see [Cormen et al., 2009, Chapter 9.3]. Thereafter, summing up all the distances takes time $O(|P|)$ per $p_j \in P$. $\qquad\square$

Nevertheless, these scores yield an improved running time by arbitrarily partitioning the estimates into $\sqrt{|P|}$ groups, finding a good estimate in every group via the truncated sum statistic each in time $|P| \cdot d$, for a total running time of $|P|^{3/2} \cdot d$, and then selecting the best estimate via another truncated sum statistic on all estimates returned by the groups in time $|P|$. The final algorithm consists of applying this idea recursively.

To prove that MINSUMSELECT outputs a good solution, we require a parameterized notion of a successful empirical mean. We say that a mean $\mu_j$ is $\gamma$-good, if $\|\mu_j - \mu\| \leq \gamma\sqrt{\frac{\varepsilon \text{OPT}}{n}}$. A straightforward application of the Chernoff bound guarantees us that all but a very small fraction of the points are $\gamma$-good. Assuming this, the following lemma determines the quality of the computed solution.

**Lemma B.3.** *Let $P$ be a set of means such that at least $\left(1 - \left(\frac{3}{10}\right)^{i+1}\right)$ of the means are $\gamma$-good. Then MINSUMSELECT$(P, i)$ returns a mean that is $5^{i+1}\gamma$-good.*

We prove this lemma by induction. We will use the following lemma in both the base case and the inductive step.

**Lemma B.4.** *Given a set of means $P$, suppose that at least $\frac{7}{10}|P|$ are $\gamma$-good. Then, the estimate returned by COMPUTEWINNER$(P)$ is $5\gamma$-good.*

*Proof.* First, let $\hat{\mu}' \in P$ be any $\gamma$-good estimator. We know that the $\frac{7}{10}|P|$-closest estimators to $\hat{\mu}'$ have distance at most $2\gamma\sqrt{\frac{\varepsilon \text{OPT}}{n}}$, which likewise implies that $\min_{\mu_j \in P} D_j \leq \frac{7}{10}|P| \cdot 2\gamma\sqrt{\frac{\varepsilon \text{OPT}}{n}}$. Now suppose that $\hat{\mu}_j$ is the estimator returned by the algorithm. Let $G(\hat{\mu}_j)$ be the set of $\gamma$-good estimators among the $\frac{7}{10}|P|$ closest to $\hat{\mu}_j$. By assumption, we have $|G(\hat{\mu}_j)| \geq \frac{4}{10}|P|$, which implies

$$\sum_{\hat{\mu}_k \in G(\hat{\mu}_j)} \|\hat{\mu}_j - \hat{\mu}_k\| \leq \frac{7}{10}|P| \cdot 2\gamma\sqrt{\frac{\varepsilon \text{OPT}}{n}} = \frac{7}{5}\gamma|P|\sqrt{\frac{\varepsilon \text{OPT}}{n}},$$

which gives

$$\min_{\hat{\mu}_k \in G(\hat{\mu}_j)} \|\hat{\mu}_j - \hat{\mu}_k\| \leq \frac{7}{2}\gamma\sqrt{\frac{\varepsilon \text{OPT}}{n}}.$$

Therefore,

$$\|\hat{\mu}_j - \mu\| \leq \min_{\hat{\mu}_k \in G(\hat{\mu}_j)} \|\hat{\mu}_j - \hat{\mu}_k\| + \|\hat{\mu}_k - \mu\| \leq \frac{7}{2}\gamma\sqrt{\frac{\varepsilon \text{OPT}}{n}} + \gamma\sqrt{\frac{\varepsilon \text{OPT}}{n}} \leq 5\gamma\sqrt{\frac{\varepsilon \text{OPT}}{n}}. \quad\square$$

*Proof of Lemma B.3.* We proceed with the induction starting from $i = 0$.

**Base Case.** For the base case $i = 0$, MINSUMSELECT$(P, 0)$ only calls COMPUTEWINNER. Thus, this case holds due to Lemma B.4.

**Inductive Step.** Let $P_1, \ldots P_{\sqrt{|P|}}$ be the clusters of $P$ computed when first calling MINSUMSELECT$(P, i)$. By assumption, we have at most $\left(\frac{3}{10}\right)^{i+1}|P|$ means that are not $\gamma$-good. This implies that the number of clusters with more than $\left(\frac{3}{10}\right)^i \sqrt{|P|}$ means that are not $\gamma$-good is at most

$$\frac{\left(\frac{3}{10}\right)^{i+1}|P|}{\left(\frac{3}{10}\right)^i \sqrt{|P|}} = \frac{3}{10}\sqrt{|P|}.$$

Denote this set by $B(P)$ and let $G(P)$ be the remaining clusters. For each $P_j \in B(P) \cup G(P)$, let $\hat{\mu}_j$ returned by MINSUMSELECT$(P_j, i-1)$. If $P_j \in G(P)$, we may use the inductive hypothesis

which states that the mean $\hat{\mu}_j$ returned by MINSUMSELECT($P_j,i-1$) is $5^i\gamma$-good. Since at least $\frac{7}{10}\sqrt{|P|}$ of the thus computed means are $5^i\gamma$-good, we may use Lemma B.4 which shows that the final mean returned by COMPUTEWINNER($\cup_{P_j\in B(P)\cup G(P)}\{\hat{\mu}_j\}$) is $5^{i+1}\gamma$-good, which concludes the proof. $\qquad\square$

To conclude the proof, we require two more arguments. First, we must show that, for an appropriate choice of $a$ and $b$, at least a $\left(1-\left(\frac{3}{10}\right)^{i+1}\right)$ fraction of the means are 1-good with probability at least $1-\delta$, allowing us to use Lemma B.3. Second, we will argue the running time. The first is a simple application of the Chernoff bound.

**Lemma B.5.** *For $a\geq 2\cdot 25^{i+1}\cdot\left(\frac{10}{3}\right)^{i+1}$ and $b=3$, at least $\left(1-\left(\frac{3}{10}\right)^{i+1}\right)$ of the means are $5^{-(i+1)}$-good with probability $1-\delta$.*

*Proof.* Let $X_i=\begin{cases}1 & \text{if }\mu_i\text{ is not }5^{-(i+1)}-\text{good}\\0 & \text{if }\mu_i\text{ is }5^{-(i+1)}-\text{good}\end{cases}$. We have $\mathbb{E}[\|\hat{\mu}_i-\mu\|^2]=\frac{1}{a}\cdot\frac{\varepsilon\text{OPT}}{n}$ due to Lemma 2.1. This implies due to Markov's inequality that $\mathbb{P}[X_i=1]\leq\frac{25^{i+1}}{a}$ which, by our choice of $a$, is less than $\frac{1}{2}\cdot\left(\frac{3}{10}\right)^{i+1}$. Thus, by the Chernoff bound

$$\mathbb{P}\left[\sum_{i=1}^{b\log\delta^{-1}}X_i\geq\left(\frac{3}{10}\right)^{i+1}b\log\delta^{-1}\right]\leq\mathbb{P}\left[\sum_{i=1}^{b\log\delta^{-1}}X_i\geq 2\cdot\frac{25^{i+1}}{a}\cdot b\log\delta^{-1}\right]$$
$$\leq\exp\left(-\frac{25^{i+1}}{3a}\cdot b\log\delta^{-1}\right),$$

which is at most $\delta$ by our choice of $b$. $\qquad\square$

The approximation guarantee now follows from Lemma B.3 and Lemma B.5. What remains to be shown is the running time. Since we split a collection of $t$ means into $\sqrt{t}$ clusters, the running time consists of the time required to recursively run the algorithm on the $\sqrt{t}$ clusters each of size $\sqrt{t}$ and consolidating via COMPUTEWINNER, which takes $O(t\cdot d)$ time. Thus, starting with an instance of size $|P_0|\in O(\log\delta^{-1})$, we can solve for the recursion

$$T(|P|)=\begin{cases}\sqrt{|P|}\cdot T(\sqrt{|P|})+|P|\cdot d & \text{if }|P|\geq|P_0|^{2^{-i}}\\|P|^2\cdot d & \text{if }|P|\leq|P_0|^{2^{-i}}\end{cases},$$

which yields the desired running time $O(i\cdot d\log\delta^{-1}+(\log\delta^{-1})^{1+2^{-i}}\cdot d)=O((\log\delta^{-1})^{1+2^{-i}}\cdot d)$.

We next formalize this and complete the proof of Theorem B.1.

*Proof of Theorem B.1.* Throughout this proof, assume $a\geq 2\cdot 25^{i+1}\left(\frac{10}{3}\right)^{i+1}$ and $b\geq 3$.

We first argue correctness, then running time. Let $P$ denote the entire set of means passed to the MINSUMSELECT algorithm. By Lemma B.3 and Lemma B.5 and with our choices of $a$ and $b$, MINSUMSELECT($P,i$) returns a 1-good mean, that is a $(1+\varepsilon)$-approximate mean with probability at least $1-\delta$.

What remains to be shown is the running time. Denote by $|P|_0=b\log\delta^{-1}$ the initial set of sample means. MINSUMSELECT($P,i$) computes $\sqrt{|P|}$ children, each of which recursively calls MINSUMSELECT. Consolidating via COMPUTEWINNER takes time $O(\sqrt{|P|}^2\cdot d)=O(|P|\cdot d)$ due to Lemma B.2. Thus the overall recursion takes time

$$T(|P|)=\begin{cases}\sqrt{|P|}\cdot T(\sqrt{|P|})+|P|\cdot d & \text{if }|P|\geq|P|_0^{2^{-i}}\\|P|^2\cdot d & \text{if }|P|\leq|P|_0^{2^{-i}}\end{cases},$$

which solves to a running time of $O(i\cdot|P|_0\cdot d+|P|_0^{1+2^{-i}}\cdot d)=O((b\log\delta^{-1})^{1+2^{-i}}\cdot d)$. The time to compute the initial set of means is $O\left(a\varepsilon^{-1}\cdot bd\log\delta^{-1}\right)$, which with our choice of $a$ and $b$, and some overestimation of the constants, leads to an overall running time of $O\left(\left(100^i\varepsilon^{-1}+\left(\log\delta^{-1}\right)^{2^{-i}}\right)\cdot d\log\delta^{-1}\right)$. $\qquad\square$

**Remark B.6.** *For any constant choice of recursion depth, the sample complexity only increases by constants. We did not attempt to optimize the constants, but the exponential dependency on $i$ is not avoidable. If we were to prioritize the running time over the sample complexity, we can set a recursion depth of $i = \log \log \log \delta^{-1}$, which achieves a running time and sample complexity of $O(\varepsilon^{-1} \log \delta^{-1} \text{poly} \left( \log \log \delta^{-1} \right) \cdot d)$.*

*Furthermore, if the recursion depth is shallow (i.e., for small values of $i$), the recursion can be improved via a better choice of cluster size. Specifically, in the case $i = 1$, that is with just one set of children, Theorem B.1 yields a running time of $O \left( (\varepsilon^{-1} + \sqrt{\log \delta^{-1}}) \cdot d \log \delta^{-1} \right)$. If we instead choose $\left( \log \delta^{-1} \right)^{2/3}$ many clusters, each consisting of $\sqrt[3]{\log \delta^{-1}}$ many estimators, we obtain a running time of $O \left( (\varepsilon^{-1} + \sqrt[3]{\log \delta^{-1}}) \cdot d \log \delta^{-1} \right)$. Similar improvements for other values $i$ are also possible, but these improvements become increasingly irrelevant compared to the bounds given in Theorem B.1 as $i$ gets larger.*

## C  Learning the Mean of a High-Dimensional Distribution

**Theorem 3.2.** *Let $X_1, X_2, \ldots, X_N$ be independent samples from distribution $\mathcal{D}$ with variance $\mu$ and covariance matrix $\Sigma$, then the coordinate-wise median-of-means estimator $\nu_{\text{CWM}}$, with probability at least $1 - \delta$, satisfies*

$$\|\nu_{\text{CWM}} - \mu\| \leq 40 \sqrt{\frac{\text{TR}(\Sigma) \log \delta^{-1}}{N}}$$

*Proof.* The $N$ samples are partitioned into $8 \log \delta^{-1}$ subsamples of size $\frac{N}{8 \log \delta^{-1}}$ each. The algorithm computes the empirical mean of each subsample and returns the coordinate-wise median of the set of empirical means.

We prove the analog of Lemma 2.3 for this case, showing that a large fraction of sample means lie close to the true mean. For an empirical mean of $\frac{N}{8 \log \delta^{-1}}$ samples, we have $\mathbb{E}[\|\hat{\mu} - \mu\|^2] = \frac{8 \text{TR}(\Sigma) \log \delta^{-1}}{N}$.

We call an empirical mean *good* if $\|\hat{\mu} - \mu\| \leq r$ where $r = 13 \sqrt{\frac{\text{TR}(\Sigma) \log \delta^{-1}}{N}}$ and let $G$ denotes the set of *good* means. Using $\text{TR}(\Sigma) = \mathbb{E}[\|X - \mu\|^2]$ and the Markov's inequality, we have $\mathbb{P}[\hat{\mu} \in G] \geq 1 - \frac{8}{13^2} \geq 0.95$. Applying the Chernoff bound, we get that with probability at least $1 - \delta$, we have $|G| \geq \frac{7}{10} \log \delta^{-1}$.

Consider $\nu_{\text{CWM}}$, the coordinate-wise median of all the sample means. For each coordinate $k$, let $L_k$ and $R_k$ be the sets of sample means such that $\hat{\mu}_{i,k} \leq \nu_{\text{CWM},k}$ and $\hat{\mu}_{i,k} \geq \nu_{\text{CWM},k}$ respectively. We know that $L_k, R_k$ have cardinality at least $\frac{b \log \delta^{-1}}{2}$. Depending on whether $\nu_{\text{CWM},k} > \mu_k$ or not, at least for one of $L_k, R_k$, we have that $|\hat{\mu}_{i,k} - \mu_k| \geq |\nu_{\text{CWM},k} - \mu_k|$.

We know that at least $\frac{7}{10}$ fraction of the means are *good* (i.e., $\|\hat{\mu}_i - \mu\| \leq r$) with probability at least $1 - \delta$. Hence, we infer that at least $\frac{7}{10} - \frac{1}{2} = \frac{1}{5}$ of the sample means are *good* and satisfy $|\hat{\mu}_{i,k} - \mu_k| \geq |\nu_{\text{CWM},k} - \mu_k|$. On average,

$$\frac{1}{|G|} \sum_{i \in G} |\hat{\mu}_{i,k} - \mu_k|^2 \geq \frac{|\nu_{\text{CWM},k} - \mu_k|^2}{5}.$$

Summing over all coordinates $k$ gives

$$\sum_{k \in [d]} \frac{1}{|G|} \sum_{i \in G} |\hat{\mu}_{i,k} - \mu_k|^2 \geq \sum_{k \in [d]} \frac{|\nu_{\text{CWM},k} - \mu_k|^2}{5} = \frac{\|\nu_{\text{CWM}} - \mu\|^2}{5}.$$

Interchanging the sums on the left-hand side, we get

$$\|\nu_{\text{CWM}} - \mu\|^2 \leq \frac{5}{|G|} \sum_{i \in G} \sum_{k \in [d]} |\hat{\mu}_{i,k} - \mu_k|^2 = \frac{5}{|G|} \sum_{i \in G} \|\hat{\mu}_i - \mu\|^2 \leq 5r^2.$$

The theorem follows from the definition of $r = 13 \sqrt{\frac{\text{TR}(\Sigma) \log \delta^{-1}}{N}}$.  □

## D   Generalization Bounds

We place our results in the context of generalization bounds for clustering problems. In this setting, we are given an arbitrary but fixed distribution $\mathcal{D}$ supported on the unit Euclidean ball $B_2^d$. The cost of a point $c$ with respect to $\mathcal{D}$ is defined as $\text{cost}_{\mathcal{D}}(c) = \int_{p \in B_2^d} \|p - c\|^2 \cdot \mathbb{P}[p] dp$. The risk is defined as $\mathcal{R} := \underset{c}{\text{argmin }} \text{cost}_{\mathcal{D}}(c)$. Given independent samples $S$ drawn from $\mathcal{D}$, we wish to compute an estimate $\hat{c}$ with cost $\hat{\mathcal{R}} = \text{cost}_{\mathcal{D}}(\hat{c})$ such that the excess risk $\hat{\mathcal{R}} - \mathcal{R}$ is minimized. The cost of a distribution $\mathcal{D}$ is in a limiting sense the average cost of a point set with all points living in $B_2^d$. Since the average cost is at most the squared radius (i.e. 1), obtaining a $(1 + \varepsilon)$ approximate solution also yields a solution that has excess risk of at most $\frac{\varepsilon \text{OPT}}{n} \leq \varepsilon$, which we then rewrite in terms of the sample size $|S|$ as $O(\frac{\log \delta^{-1}}{|S|})$. This discussion is summarized in the following corollary.

**Corollary D.1.** *Given a set of independent samples $S$ drawn from some underlying arbitrary but fixed distribution supported on $B_2^d$, the geometric median-of-means estimator has an excess risk for the least squared distances objective of*

$$\hat{\mathcal{R}} - \mathcal{R} \leq \gamma \cdot \frac{\log \delta^{-1}}{|S|}$$

*with probability $1 - \delta$ for some absolute constant $\gamma > 0$.*

Note that the learning rate given by Corollary D.1 stands in contrast to learning rates that are achievable for other center-based problems such as indeed the geometric median with objective

$$\text{cost}_{\mathcal{D}}(c) = \int_{p \in B_2^d} \|p - c\| \cdot \mathbb{P}[p] dp,$$

or for the $k$-means problem with objective

$$\text{cost}_{\mathcal{D}}(C) = \int_{p \in B_2^d} \min_{c \in C} \|p - c\|^2 \cdot \mathbb{P}[p] dp$$

for a $k$-center set $C$. As mentioned in the related work section, all of these objectives require learning rates of at least $\sqrt{\frac{1}{|S|}}$, ignoring problem specific parameters.

## E   Lower Bounds

We complement our algorithmic results with matching lower bounds.

### E.1   A High Probability Lower Bound for Mean-Estimation

**Theorem E.1.** *Any sublinear algorithm that outputs a $(1 + \varepsilon)$-approximate Euclidean mean with probability at least $1 - \delta$ must sample at least $\Omega(\varepsilon^{-1} \log \delta^{-1})$ many points.*

The idea is to generate two instances that a sublinear algorithm for approximating means has to distinguish between and then bound the number of samples required to distinguish between such distributions. The first instance is virtually identical to the one used by Cohen-Addad et al. [2021]. It places $n$ points at 0 and $\varepsilon n$ points at 1. Thus, the optimal mean is $\frac{\varepsilon}{1+\varepsilon}$. A routine calculation via Lemma 2.2 shows that the optimal cost is $\frac{\varepsilon n}{1+\varepsilon}$. The second instance places all points at 0. Since 0 is not a sufficiently good approximate mean for the first instance, a sublinear algorithm can distinguish between the two instances, which yields a lower bound on the sample complexity.

The two instances lie within the unit Euclidean ball. By normalizing the number of points, it also yields a distribution supported on the unit Euclidean ball, which implies that the generalization bounds given in Corollary D.1 are sharp.

*Proof.* We give two instances. The first instance places $n$ points at 0 and $\varepsilon n$ points at 1. Thus, the optimal mean is placed at $\frac{\varepsilon}{1+\varepsilon}$. A routine calculation via Lemma 2.2 shows that the optimal cost is $\text{OPT} = \frac{\varepsilon n}{1+\varepsilon}$. The second instance places all points at 0.

First we argue that a sublinear algorithm can distinguish between these two instances. Any approximate mean for the second instance must output $0$. If we output $0$ for the first instance, we incur a cost of $\varepsilon n = \frac{\varepsilon n(1+\varepsilon)}{1+\varepsilon} = \text{OPT} + \varepsilon \cdot \text{OPT}$, hence any algorithm improving over a $(1+\varepsilon)$ approximation for the former cannot output $0$. Thus, a necessary condition to distinguish between the two instances is for the algorithm to sample at least one point at $1$.

Suppose we sample $m$ points. Our goal is to show that for $m \in \Omega(\varepsilon^{-1} \log \delta^{-1})$, the probability that all of the sampled points are drawn from $0$ is less than $\delta$ for the first instance. Let $X_i$ denote the indicator variable of the $i$th sampled point. We have

$$\delta \geq \mathbb{P}[\forall \, i, \ X_i = 0] = \mathbb{P}[X_1 = 0]^m = \left( \frac{1}{1+\varepsilon} \right)^m$$

$$= \exp\left( m \ln \frac{1}{1+\varepsilon} \right) = \exp\left( -m \ln(1+\varepsilon) \right)$$

$$> \exp\left( -m\varepsilon \right),$$

where the final inequality follows from the Mercator series $\ln(1+\varepsilon) = \sum_{i=1}^{\infty} \left( \frac{\varepsilon^i}{i} \right) \cdot (-1)^{i+1}$. Thus, for any $m < \varepsilon^{-1} \cdot \log \delta^{-1}$, we do not achieve the desired failure probability. Conversely, $m \in \Omega(\varepsilon^{-1} \log \delta^{-1})$ for an algorithm to succeed. $\qquad\square$

## E.2 The Empirical Mean is not a Good Estimator

We briefly show that the arguably most natural algorithm that outputs the empirical mean of a subsampled point set has a substantial increase in sample complexity in the high success probability regime and therefore also a substantial increase in running time compared to the algorithms presented in this paper. The result is folklore, but included for completeness.

**Theorem E.2.** *For all $\varepsilon, \delta \in (0,1)$, there exists an instance such that $\Omega(\varepsilon^{-1}\delta^{-1})$ independently sampled points are required for the empirical mean to be a $(1+\varepsilon)$-approximate Euclidean mean with probability at least $1 - \delta$.*

The proof is based on reducing the problem of finding a good approximate mean with high probability to giving an example distribution for which the Chebychev inequality is tight.

*Proof.* Consider a sample $|S|$, with $|S|$ being specified later. We generate an instance as follows. We place a $\frac{1}{2|S|^2\varepsilon}$-fraction of the points each at $-|S|\sqrt{\varepsilon}$ and $|S|\sqrt{\varepsilon}$ and the remaining $1 - \frac{1}{|S|^2\varepsilon}$ fraction at $0$. Then the optimal solution places the mean at $0$ and has average cost $(|S|\sqrt{\varepsilon})^2 \cdot \frac{1}{|S|^2\varepsilon} = 1$. By Lemma 2.2, this implies that the empirical mean $\hat{\mu} = \frac{1}{|S|} \sum_{p \in S} p$ is a better than $(1+\varepsilon)$-approximate mean if and only if $|\hat{\mu}| < \sqrt{\varepsilon}$. We set the failure probability, that is the probability $\mathbb{P}[|\hat{\mu}| \geq \sqrt{\varepsilon}] = \delta$. Then $\mathbb{P}[|\hat{\mu}| \geq \sqrt{\varepsilon}]$ is at least the probability that we sample exactly one point from either $-|S|\sqrt{\varepsilon}$ or $|S|\sqrt{\varepsilon}$ and the remaining $|S| - 1$ points from $0$. Using Bernoulli's inequality and the density function of the binomial distribution, we therefore have

$$\delta = \mathbb{P}[|\hat{\mu}| \geq \sqrt{\varepsilon}] \geq |S| \cdot \frac{1}{|S|^2\varepsilon} \cdot \left( 1 - \frac{1}{|S|^2\varepsilon} \right)^{|S|-1} \geq \frac{1}{|S|\varepsilon} \cdot \left( 1 - \frac{1}{|S|\varepsilon} \right).$$

Solving $\frac{1}{|S|\varepsilon} \cdot \left( 1 - \frac{1}{|S|\varepsilon} \right) \leq \delta$ for $|S|$ then yields $|S| \in \Omega(\varepsilon^{-1}\delta^{-1})$, as desired. $\qquad\square$

Although the empirical mean is the fastest among all mean estimators (it can be computed in closed form), the above construction shows that it is not a robust estimator of the true mean—even in one dimension. Indeed, for heavy-tailed distributions, the best available tail bound is Chebyshev's inequality, which is exponentially weaker. In fact, Catoni [2012] (similarly to the construction above) shows that there exist distributions for which Chebyshev's inequality is tight. In contrast, the empirical mean performs well when estimating the mean of subgaussian distributions, where the associated tail bounds are sharply exponential. In practice, data distributions are often not as adversarial as those constructed to demonstrate the failure of the empirical mean (e.g., Catoni [2012]). This explains why the empirical mean is frequently observed to perform well in real-world scenarios, both in terms of computational efficiency and estimation accuracy.

# F    Comparison of Geometric Median and the Coordinate-wise median

In this section, we give two instances which demonstrate that neither of the coordinate-wise median or the geometric median is strictly better than the other for the problem of mean-estimation.

**Proposition F.1.** *Let $\delta > 0$ be fixed. There exist instances such that, with probability at $1 - O(\delta)$,*

*(1) The coordinate-wise median of $K = \frac{\log \delta^{-1}}{2\left(\frac{1}{2} - e^{-1}\right)^2}$ sample means of $\varepsilon^{-1}$ sampled points coincides with the true mean, but the geometric median of the same $K$ sample means does not;*

*(2) The geometric median of $\log \delta^{-1}$ sample means of $\varepsilon^{-1}$ sampled points is a better approximation to the true mean than the coordinate-wise median is.*

*Proof.* For part (1), we construct an instance for which the coordinate-wise median is a better estimator than the geometric median. Consider the uniform distribution on the $d$-dimensional simplex where $d = O(\varepsilon^{-2}\delta^{-2}\log^2 \delta^{-1})$. The value of $d$ ensures that with probability at least $1 - \delta$, each of the $\varepsilon^{-1}\log \delta^{-1}$ vertices sampled are distinct. We show that in this case the coordinate-wise median is a better aggregation procedure than the geometric median. The coordinate median $\nu_{\mathrm{CWM}}$ at the origin, while the true mean is $\mu = (1/d, 1/d, \ldots, 1/d)$. The geometric median can be shown to lie at the empirical mean of all the samples due to symmetry, which is $\mu_{\mathrm{GM}} = (\varepsilon/\log(\delta^{-1}), \varepsilon/\log(\delta^{-1}), \ldots, 0, \ldots, 0)$ where there are exactly $\log(\delta^{-1})/\varepsilon$ non-zero coordinates. Comparing the distance, we see that $\|\mu - \nu_{\mathrm{CWM}}\| \leq \|\mu - \mu_{\mathrm{GM}}\|$. To conclude, we note that with probability at least $1 - \delta$, the coordinate-wise median is a better estimator than the geometric median.

For part (2), we wish to prove that there exists an instance in which the geometric median is better than the coordinate-wise median. Consider the uniform distribution on the $d$-dimensional simplex, where $d = c\varepsilon^{-1}$. We choose $c$ to be a constant large enough so that in every subsample, the probability of a vertex sampled is $\frac{1}{3}$ (we only require it to be bounded away from $\frac{1}{2}$). As in the last example, we have that the actual mean is $(1/d, 1/d, \ldots, 1/d)$. We observe that the coordinate-wise median $\nu_{\mathrm{CWM}}$ is the origin with high probability. While as $\delta \to 0$, the geometric median tends towards true mean $\mu$. $\square$

# G    Further Experimental Evaluation

We present here further experimental evaluation and specifically we plot the accuracy vs. sample size and runtime vs. sample size, as well as the accuracy vs. dimension and runtime vs. dimension.

We first describe some summary statistics for the datasets we consider:

| Dataset | Shape | Mean | Std-Dev | Min | Max |
|---|---|---|---|---|---|
| MNIST | (60,000; 784) | 0.1306 | 0.3081 | 0.0000 | 1.0000 |
| Fashion-MNIST | (60,000; 784) | 0.2860 | 0.3530 | 0.0000 | 1.0000 |
| CoverType | (581,012; 54) | 0.4567 | 0.4981 | 0.0000 | 1.0000 |

Table 1: Summary statistics for datasets.

**Accuracy and Runtime vs. Sample Size.**    We plot how accuracy and runtime vary as a function of the sample size. First, we plot this relationship in log-scale (Figures 4-5).

As they visually compress differences and obscure linear growth, especially when the slope is small, we replot accuracy vs. sample size in linear scale (Figure 6). The trends are indeed mildly increasing and consistent with near-linear growth, albeit with a small slope. We believe the soft growth observed on the original dataset is largely due to implementation-level factors. In particular, our use of NumPy's vectorized operations can lead to non-obvious runtime behavior, as such operations benefit from low-level optimizations (e.g., memory locality, multi-threaded backends) and often incur fixed overheads.

We also add further experimentation for larger sample sizes (Figure 7). Specifically, we generated a synthetic dataset from a 200-dimensional multivariate Gaussian distribution and extended the sample

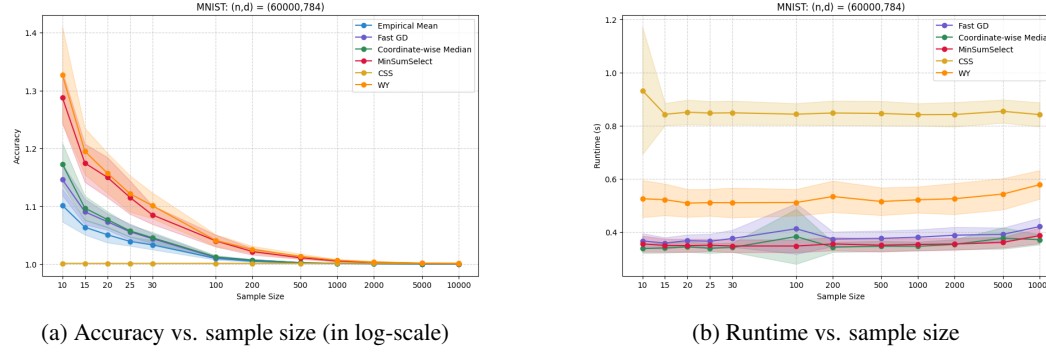

(a) Accuracy vs. sample size (in log-scale)  (b) Runtime vs. sample size

Figure 4: MNIST Dataset: Accuracy and runtime against sample size.

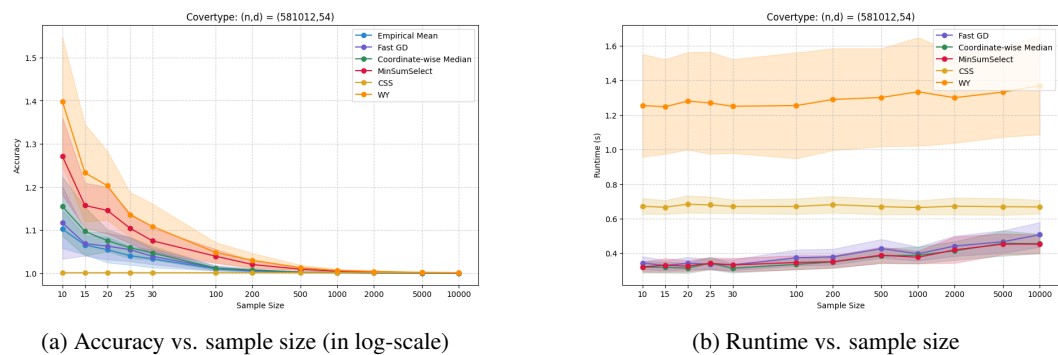

(a) Accuracy vs. sample size (in log-scale)  (b) Runtime vs. sample size

Figure 5: CoverType Dataset: Accuracy and runtime against sample size.

size up to $5 \cdot 10^5$. The results are in line with theory: They show that the error decreases sharply as the sample size grows and that the runtime grows linearly with sample size. Unfortunately, we were unable to go beyond this sample size due to memory limitations in our environment, where the process exhausts the available RAM.

**Accuracy and Runtime vs. Dimension.** The experiments here illustrate how accuracy and runtime vary with dimensionality (Figure 8). Specifically, we generated a synthetic dataset of $10,000$ samples drawn from multivariate Gaussian distributions with dimensions ranging from 10 to 1000. As predicted by theory, the errors of both the coordinate-wise median and the fast gradient descent estimator remain low and stable across dimensions. Interestingly, the CSS algorithm shows rapid improvement as the dimension increases, stabilizing around dimension $500$. As expected, the empirical mean achieves the best accuracy since the data are drawn i.i.d. from a Gaussian distribution. In terms of runtime, we observe a roughly linear growth with dimension, in line with theory.

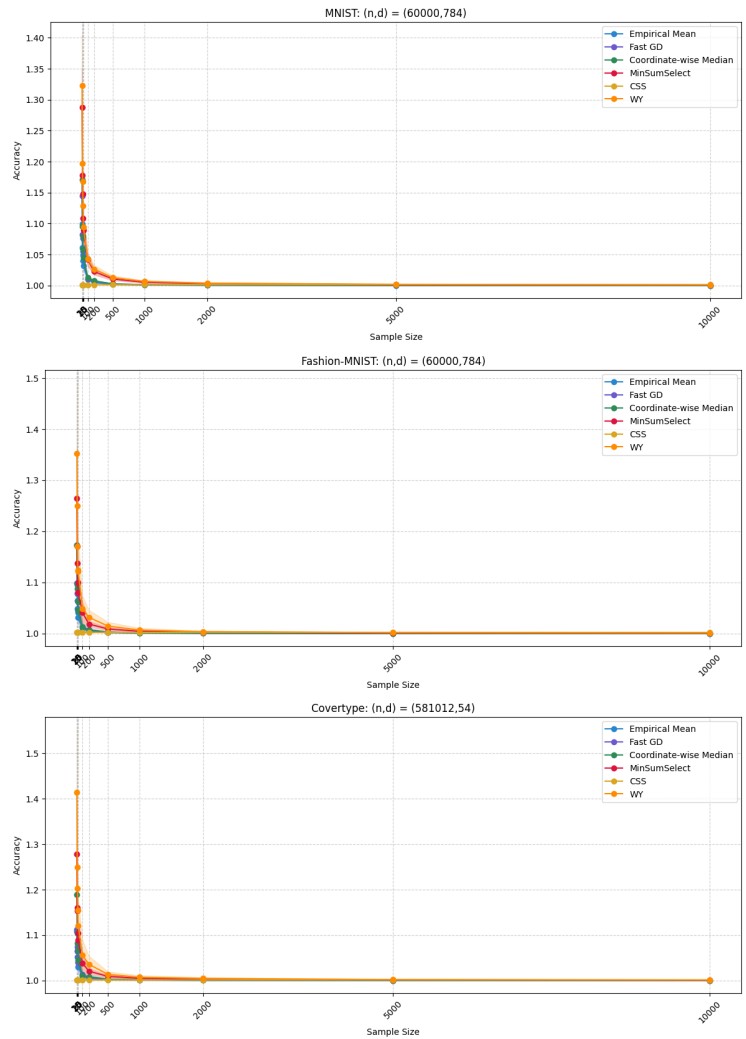

Figure 6: MNIST, Fashion-MNIST and CoverType Datasets: Accuracy against sample size (in linear scale).

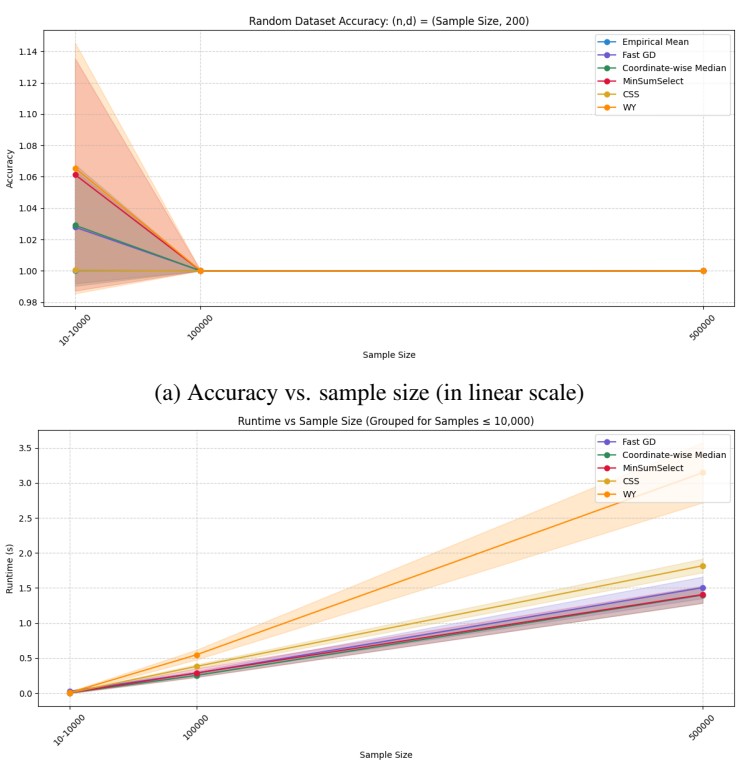

(a) Accuracy vs. sample size (in linear scale)

(b) Runtime vs. sample size

Figure 7: Synthetic Dataset: Accuracy and runtime against sample size (in linear scale).

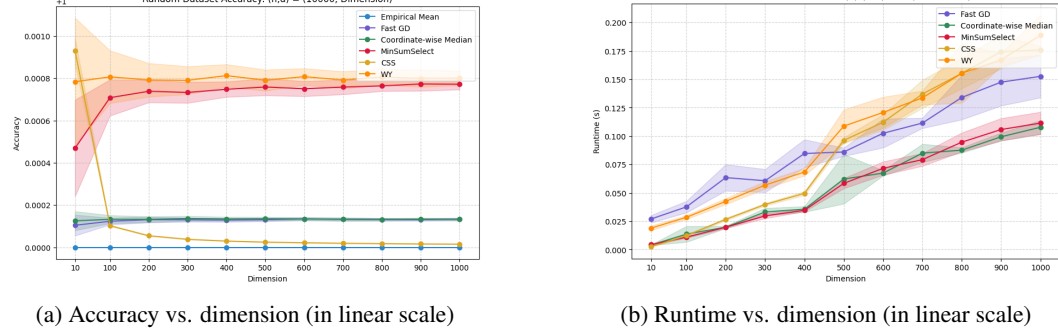

(a) Accuracy vs. dimension (in linear scale)

(b) Runtime vs. dimension (in linear scale)

Figure 8: Synthetic Dataset: Accuracy and runtime against dimension.

