# OpenReview forum: "Simple and Optimal Sublinear Algorithms for Mean Estimation"
_NeurIPS.cc/2025/Conference — NeurIPS 2025 poster_

### Official Review · Reviewer_sRHN · 2025-06-20

**Clarity:** 4
**Significance:** 4
**Originality:** 3
**Rating:** 4
**Confidence:** 3

**Summary:**

The paper discusses sublinear mean estimation and presents three different algorithms. For each of these algorithms, running time analysis and optimality discussions are provided. Several experiments comparing proposed algorithms with existing algorithms are provided.

**Questions:**

Majority of my questions are about the proofs and experiments.

Q1. In line 143, it is stated that earlier values of constants were set as $a = 1440, b = 50$, but neither of these appear to be discussed in detail prior to this line. Is some text missing or can some more context be shared as to why these constants were chosen?

Q2. Overall I found the proof of Lemma 2.3 to be quite vague and would prefer it to be explained in more detail, particularly because it is one of the cornerstone results of the paper. For instance, why do we get $\|\hat{\mu}_i-\mu\|^2 > r^2$ with probability at most 0.1, and where does this 0.1 come from? How does $-2/81$ appear in line 145-146?

Q3. In line 160 it is stated that "$L_k, R_k$ have cardinality at least $b\log \delta^{-1}/2$." What does this mean when $L_k + R_k = b\log \delta^{-1}$? Following this logic, can you explain in a bit more detail how we can infer at least $1/5$ of the samples satisfy $|\hat{\mu}_{i,k}-\mu_k|\geq |\nu_k - \mu_k|$ from this?

Q4. Following Q3, can you clarify the logic in lines 160-161? In particularly, since $L_k, R_k$ are sets, the sentence "at least for one of $L_k,R_k$, we have $|\hat{\mu}_{i,k}-\mu_k|\geq |\nu_k - \mu_k|$" doesn't make a lot of sense.

Q5. Following Q4., because of Q4 it's quite difficult to judge if the sentence "Depending on whether $\nu_k>\mu_k$ or not, at least for one of $L_k,R_k$, we have that $|\hat{\mu}_{i,k}-\mu_k|\geq |\nu_k - \mu_k|$." is logical. Wonder if you can help clarify it with some examples, for example say if $\mu_k = 0, \hat{\mu}_k = 2, \nu_k = 3$. In this case, the left hand side is 2 while the right hand side is 3 and the inequality doesn't hold.

Q6. As briefly mentioned in an earlier section of the review, it appears from the experiments that the empirical mean performs consistently better than proposed algorithms, and is likely fast as well; can you share more insights on how this may be happening?

Q7. Wonder if there are more experiments that look at how the algorithms perform under different data dimension, say, varying between 1- 1000 instead of a fixed value like 784 or 54.

Additional comments regarding formatting and typos:

1. Line 65: "three algorithm"
2. Line 109: "studided"
3. Line 135: "appoximation"
4. Line 191: "proof for the our setting"
5. Line 313: "runing time"
6. Line 330: "these observation"

**Ethical Concerns:**

["NO or VERY MINOR ethics concerns only"]

**Final Justification:**

My original concerns about details of the proofs were addressed sufficiently. Overall I think the paper has good quality and it will improve with further revisions. After engaging with other reviewers, I agree about some concerns regarding novelty. Based on this I'm suggesting a score of 4.

**Limitations:**

The authors adequately addressed limitations regarding uniform sampling and data corruption.

**Paper Formatting Concerns:**

N/A.

**Quality:**

3

**Strengths And Weaknesses:**

1. Perhaps most importantly, the new proposed algorithms are optimal in many aspects such as scalability and efficiency, which is arguably the highlight of the paper. There are some follow up questions regarding proof details which are deferred to the "Questions" section of the review, but if shown to be correct, the results are fairly interesting.
2. Overall, the paper is well-organized and discusses background, key concepts, and contributions well. It is clear that the authors spent a lot of effort polishing the paper. Several suggestions regarding typos or formatting issues are deferred to a later section.
3. The experiments are relatively weak. For instance, it is clear that the empirical mean consistently outperforms proposed algorithms in terms of accuracy. While the empirical mean is not included in *runtime* comparisons, it should be significantly faster than the proposed algorithms, if I understand correctly. Perhaps some additional experiments or discussion would be helpful to address these observations.
4. Following point 3., I think the result of Theorem 3.2 regarding learning the mean of high-dimensional distributions is also quite interesting but there doesn't appear to be any experiments that support the dimension-free result. Might be helpful if some extra experiments or discussions can be made there.

---

> ### Author Rebuttal · Authors · 2025-07-31
>
> We thank the reviewer for the helpful feedback. As per NeurIPS 2025 policy, we cannot include links or images in the rebuttal or repository. We’ve therefore provided tables (in earlier responses) with the same information as the plots and appreciate your understanding given these limitations.
>
> **Regarding proofs.**
>
> **Q1 Response.** We apologize for the confusion. The sentence has a typo and should instead be "We set the values of the constants to be \dots ". In the answer to the next question, please find a detailed response including motivations behind the values of the constants.
>
> **Q2 Response.** For the algorithms, the first step is to sample $b \log \delta^{-1}$ subsamples of size $a \varepsilon^{-1}$ each and compute their means. We want our final estimate to be within $\|\hat \mu - \mu \| \leq \sqrt{\frac{\varepsilon OPT}{n}}$ distance of the true mean, as it implies a $(1+\varepsilon)$-approximation. Define $r = \frac{1}{11} \sqrt { \frac{\varepsilon OPT}{n}}$ and  we call a mean \emph{good} if $\|\hat \mu - \mu\| \leq r$. Further, we define the good event $\mathcal{E}$ to be when at least $7/10$ fraction of the means are \emph{good}. The value $7/10$ is chosen to an arbitrary constant strictly greater than $1/2$.
>
> **Lemma 2.3.** Event $\mathcal{E}$ holds with probability at least $1-\delta$.
>
> The proof contains the following steps:
> 1. A sample mean derived from $1440 \varepsilon^{-1}$ samples is \emph{good} with probability $0.9$.
> 2. If we have $50 \log \delta^{-1}$ subsamples of $1440 \varepsilon^{-1}$ points each, at least $7/10$ fraction of the mean will be \emph{good} with probability $1-\delta$.
> We now give a detailed version of the proof with an explanation of the constants as well. For now, the values of constants are set as follows - $a = 1440, b = 50$.
>
> **Proof.** From Lemma 2.1 in the paper, we know that $ \mathbb{E} [ \| \mu - \hat \mu \|^2 ]= \frac{1}{|S|} \frac{OPT}{n} = \frac{1}{a \log \varepsilon^{-1}} \frac{OPT}{n}$. The Markov's inequality can be used to bound the probability that a single mean is \emph{not good} - $\mathbb{P}[\| \mu - \hat \mu \|^2 > r^2] \leq \frac{r^2}{\mathbb{E} [ \| \mu - \hat \mu \|^2 ]} \leq 0.1$. Conversely, the probability that a single mean is \emph{good} is at least 0.9. We require this probability to be strictly greater that $0.7$ in order to be able to say that the good event $\mathcal{E}$ holds with high probability. Specifically, the value of $a$ is set to $1440$ to achieve this.
>
> Next, we use the (multiplicative) Chernoff bound to show that if we have $b \log \delta^{-1}$ means, then with high probability, at least 0.7 fraction of the means are \emph{good}.
>
> The Chernoff bound says $\mathbb{P}[X < (1 - \theta)\mu] \leq \text{exp}(-\theta^2\mu/2)$, where $X$ is the sum of independent random variables and the expected value $\mu = \mathbb{E}[X]$. Setting $\mu \geq 0.9 b \log \delta^{-1}$, $\theta = 2/9$, we get
>
> $\mathbb{P}[|G| \leq 0.7 b \log \delta^{-1}] \leq \text{exp}(-\frac{2}{81} b \log \delta^{-1}).$
>
> Thus, the constant $-2/81$ is the result of applying the Chernoff bound by setting the parameters appropriately. We choose the value of $b$ so that $-(2b)/81 < -1$ and the probability bound achieved is at most $\delta$. Therefore, we set $b=50$.
>
> We will add a more detailed version of the lemma and its proof to the final version of the paper.
>
> **Q3 Response.** We have $|L_k \cup R_k| = b \log \delta^{-1}$ which is the total number of sample means. Recall that $L_k$ consists of the sample means such that $\hat \mu_k \leq \nu_{CWM,k}$. And $R_k$ is the set of sample means such that $\hat \mu_k \geq \nu_{CWM,k}$. The key detail is that there are points (such that $\hat \mu_k = \nu_{CWM,k}$) which belong to both. Along with the definition of the coordinate-wise median, we have $|L_k|, |R_k| \geq b \log \delta^{-1}/2$. Note that if there are no points such that $\hat \mu_k = \nu_{CWM,k}$, then $|L_k|= |R_k| = b \log \delta^{-1}/2$.
>
> We provide a detailed case analysis as to why at least $1/5$ fraction of the samples are good and satisfy $|\hat \mu_{i,k} - \mu_k| \ge |\nu_{CWM,k} - \mu_k|$.
>
> First, we note that when the good event $\mathcal{E}$ holds,  at least $1/5$ fraction of the means are *good* in both the sets $L_k$ and $R_k$. This follows from the fact that at least $7/10$ fraction of the means are good and the number of good means in $L_k$/$R_k$ is at least $7/10 - 1/2 = 1/5$.
>
> **Case I:** $\mu_k \leq \nu_{CWM,k}$. In this case, we have for all means in $R_k$, $|\hat \mu_{i,k} - \mu_k| \ge |\nu_{CWM,k} - \mu_k|$.
>
> **Case II.** $\mu_k > \nu_{CWM,k}$. In this case, we have for all means in $L_k$, $|\hat \mu_{i,k} - \mu_k| \ge |\nu_{CWM,k} - \mu_k|$.
>
> From the above statements, we have for at least one of $L_k$ and $R_k$, we have that at least $1/5$ fraction of the means are good and satisfy $|\hat \mu_{i,k} - \mu_k| \ge |\nu_{CWM,k} - \mu_k|$.
>
> **Q4 Response.** We apologize for the typo. We want to say that the claim is true for all means in $L_k$ or $R_k$. The formal statement is as follows. At least one of the following statements is true:
> 1.  For all $\hat \mu$ in $L_k$, we have that $|\hat \mu_{i,k} - \mu_k| \ge |\nu_{CWM,k} - \mu_k|$.
> 2. For all $\hat \mu$ in $R_k$, we have that $|\hat \mu_{i,k} - \mu_k| \ge |\nu_{CWM,k} - \mu_k|$.
>
> We will update the final version of the paper accordingly.
>
> **Q5 Response.** Please refer to our answer to Q3 for the detailed proof. In the example given by the reviewer, $\nu_{CWM,k} = 3$, implies that there are equal number of points with $\hat \mu_k < 3$ and $\hat \mu_k > 3$. So, if there exists a sample mean such that $\hat \mu_k = 2 < 3$, then there will also exist at least one mean such that $\hat \mu_k > 3$. And for that mean, which is in $R_k$, we have $|\hat \mu_{i,k} - \mu_k| \ge |\nu_{CWM,k} - \mu_k|$.
>
> **Regarding experiments.**
>
> **Q6 Response.** As the reviewer correctly notes, the empirical mean is the fastest among all mean estimators, as it can be computed in closed form. Despite this computational advantage, it is not a robust estimator of the true mean—even in one dimension. That said, the empirical mean performs well when estimating the mean of subgaussian distributions, where the associated tail bounds are sharply exponential. In contrast, for heavy-tailed distributions, the best available tail bound is Chebyshev’s inequality, which is exponentially weaker. In fact, Catoni (2012) shows that there exist distributions for which Chebyshev’s inequality is tight. Nevertheless, in practice, data distributions are often not as adversarial as those constructed to demonstrate the failure of the empirical mean (e.g., Catoni (2012)). This explains why the empirical mean is frequently observed to perform well in real-world scenarios, both in terms of computational efficiency and estimation accuracy.
>
> **Q7 Response.** We thank the reviewer for the suggestion. We have run two experiments: They illustrate how accuracy and runtime vary with dimensionality. Specifically, we generated a synthetic dataset of 10,000 samples drawn from multivariate Gaussian distributions with dimensions ranging from 10 to 1000. As predicted by theory, the errors of both the coordinate-wise median and the fast gradient descent estimator remain low and stable across dimensions. Interestingly, the CSS algorithm shows rapid improvement as the dimension increases, stabilizing around dimension 500. As expected, the empirical mean achieves the best accuracy since the data are drawn i.i.d. from a Gaussian distribution. In terms of runtime, we observe a roughly linear growth with dimension, in line with theory. Unfortunately, we could not insert these tables due to characters constraints, but we will add these new experimental results (and the corresponding plots) to the final version of the paper.

---

> > ### Comment · Reviewer_sRHN · 2025-08-04
> > **Thank you for the response**
> >
> > Thank you for the detailed responses to my questions. Most of my concerns were addressed. I will update my review and final justification accordingly.

---

### Official Review · Reviewer_Vydg · 2025-06-28

**Clarity:** 3
**Significance:** 2
**Originality:** 2
**Rating:** 3
**Confidence:** 3

**Summary:**

The paper considers the following question: given $n$ points in the $d$-dimensional Euclidean space, we want to find a $(1+\varepsilon)$-approximation algorithm for finding the mean (where the objective is the $1$-means cost). The goal is to randomly samply certain points and use a fast algorithm to find the approximate mean (with probability at least $1-\delta$). There were known lower bounds implying lower bound of $\Omega(1/\varepsilon \cdot \log(1/\delta))$. There are almost matching upper bounds (with an extra log factor). The paper gives a simple algorithm which samples $\Omega(1/\varepsilon \cdot \log(1/\delta))$ points and uses the well-known median of means trick to show that this gives an optimal sample complexity bound. They also give a gradient descent based algorithm which seems to work well on real data sets.

**Questions:**

1. What is novel in techniques here? The median of means seems to be a standard idea.
2. What about $\ell_p$ norms in general? Do the ideas here extend to these more general versions?

**Ethical Concerns:**

["NO or VERY MINOR ethics concerns only"]

**Limitations:**

yes.

**Quality:**

3

**Strengths And Weaknesses:**

Strengths:

1. Simple algorithm achieving optimal sample complexity bound for the 1-means problem.
2. Validation of algorithms by experiments.

Weakness:

1. Technique is well known and has been applied to many related problems in streaming.
2. Prior works gave almost optimal sample complexity (upto log error) for all $\ell_p$ norms, whereas the idea here works only for $\ell_2$ norm.
3. Improvement over prior work is only $\log(1/\varepsilon)$.

---

> ### Author Rebuttal · Authors · 2025-07-30
>
> We appreciate the reviewer’s comments and the opportunity to clarify the contributions and novelty of our work. We respond to the raised questions and comments in a unified manner.
>
> Our primary goal is to achieve optimal runtime in addition to optimal sample complexity for robust mean estimation in streaming settings. While prior works have indeed addressed sample complexity (often for all $\ell_p$ norms), their algorithms are generally not optimized for runtime. In contrast, our results focus specifically on the $\ell_2$ norm because our runtime improvements rely on structural properties and lemmas (e.g., Lemmas 2.1 and 2.2) that are unique to this norm. Extending such runtime guarantees to general $\ell_p$ norms remains an interesting open direction.
>
> Regarding the techniques: although median-of-means is a classical idea in robust estimation, our contributions lie in how we combine and extend known techniques to obtain new algorithmic and theoretical insights.
> The median-of-means estimator is designed for the one-dimensional mean estimation problem. Extending the idea to higher dimensions is non-trivial. There is no standard notion of a median
> for multivariate data, and it is not clear a priori what definition of multivariate
> median generalizes the median-of-means estimator for one-dimension.
>
> Specifically, our contributions are:
>
> - We introduce a gradient-descent-based algorithm that starts from the coordinate-wise median-of-means and efficiently converges to the true mean. This process provably runs in linear time, and to the best of our knowledge, this is the first such demonstration in this setting.
>
> - We show that the coordinate-wise median-of-means achieves dimension-free sample complexity in a distributional mean estimation setting—an aspect that, to our knowledge, was overlooked in prior literature.
>
> - In addition, we propose a novel estimator based purely on order statistics, which offers a distinct alternative for mean estimation and runs in near-optimal time.
>
> As for the claimed improvement: there is also an improvement in by the order of $\log \delta^{-1}$ in both sample complexity and running time. While the improvement in terms of the precision $\varepsilon$ may be more modest (even if it is optimal now and was not before), it can be argued that we obtain a quadratic improvement in both running time and sample complexity in terms of the failure probability. Moreover, breaking the $\log^2(1/\delta)$ barrier has been a notable challenge in the literature, and our method addresses this without sacrificing either sample complexity or computational efficiency in other parameters.

---

> > ### Comment · Reviewer_Vydg · 2025-08-05
> >
> > Thanks for your rebuttal. I see your point, but (i) when you say that " There is no standard notion of a median for multivariate data", I thought Theorem 3.1 is just doing this for each coordinate independently, and the proof of Theorem 3.1 is again quite standard. (ii) you say " coordinate-wise median-of-means achieves dimension-free sample complexity"...but doesn't the error depend on the dimension? Trace of Sigma matrix would be dimension dependent. So if you want to make error tiny, one needs dimension dependent sample complexity.

---

> > > ### Author Response · Authors · 2025-08-05
> > >
> > > We thank the reviewer for a quick response.
> > > (i) We would like to reiterate the fact that though the proof is standard/elementary, previous texts on mean estimation of high-dimensional distribution seem to have missed it. Please refer to survey by Lugosi and Mendelson and "Geometric median and robust estimation in banach spaces" by Minsker for further details.
> > >
> > > (ii) Our proof does show a dimension-free bound of the sample complexity, as it does not depend on the dimension explicitly. The lower bound for the sample complexity is given by $\sqrt{\frac{\text{Tr}(\Sigma)}{N}}$ even for the 1-dimensional case, so that is unavoidable.
> > > Note that value of $\text{Tr}(\Sigma)$ might not increase with the dimension $d$ (even though it might). Previous bounds included a factor $\log d$. For a contrast, consider two distributions with the same $\text{Tr}(\Sigma)$, one in $\mathbb{R}^1$ and the second in $\mathbb{R}^{1000}$. Our bounds show the same upper bound for error rate of both the distributions, while the prior results give a worse bound for the second one. Our contribution is to say that there is no additional dependence on the dimension.

---

> > > > ### Comment · Reviewer_Vydg · 2025-08-07
> > > >
> > > > Thanks for your response.

---

### Official Review · Reviewer_1w8V · 2025-07-02

**Clarity:** 3
**Significance:** 2
**Originality:** 3
**Rating:** 4
**Confidence:** 3

**Summary:**

The authors present several novel methods to estimate a multivariate mean of a ground data, including theoretical proofs for optimality of sampling complexity. The methods are built as variants of of an algorithm that computes several sample means and is parametrized by an aggregation function for said means. The authors prove theoretical guarantees of approximation optimality and asymptotic relative runtime optimality of those methods, and evaluate them against MNIST/F-MNIST datasets, comparing to other baseline methods.

**Questions:**

I believe the clarity of Lemma 2.3 should be adjusted, as per discussion before.

While this is a more theoretical paper (or even more importantly because of that), I believe an experimentation on synthetic data could be crucial here. In the simplest case, the same experiments from MNIST could be run on a sample from e.g. a Gaussian distribution, perhaps also testing limits with much larger sample sizes.

One question I have about Algorithm 2: is there any statement about algorithm's asymptotic convergence to true geometric median (in relation to T)? Unless I missed it, perhaps it is good to clarify whether it's an open question, or perhaps it is easy to construct a set of points that has a unique geometric median, but the method does not go infinitesimally close to it.

I believe addressing these issues would bring the paper to a higher quality.

**Ethical Concerns:**

["NO or VERY MINOR ethics concerns only"]

**Final Justification:**

I believe the additional experiments, clarification to existing figures and the addition of some missing definitions improve the quality of the paper.

Based on authors' reply and how I imagine the new version, I readjust my grading to "Borderline Accept". For a full "Accept" rating, one would have to verify the final version of the paper (which is currently not available due to review process restrictions) with updated plots to also reevaluate the significance/quality of the experimental results. The rest of the paper is solid.

**Limitations:**

Yes.

**Paper Formatting Concerns:**

No formatting concerns.

**Quality:**

3

**Strengths And Weaknesses:**

The paper is written in a very clear English and good formulations. The authors strive for theoretical soundness, and for the major part provide thorough theoretical justification for their claims.

While the methods, as the paper's title suggest, are simple, this adds more clarity to their analysis. Moreover, the fast gradient method described for one of the algorithms, that essentially replaces a gradient step with a calculation of one-dimensional median along the gradient direction, is a rather neat idea.

Perhaps because of limited space, some proofs in the main paper feel too condensed and hop over several steps, reducing the clarity. One particular place is the beginning of Lemma 2.3's proof that starts with the phrase "Recall that the values of the constants were set as a = 1440, b = 50". In all fairness, the values a and b were barely introduced; they first mentioned inline in Algorithm 1, and it's trivial how they appear from the big-O of sample sizes. However neither these particular values 1440 and 50 are mentioned before, nor is it clear how exactly they are used.

Another example of a jump in the proof, to me, is in line 164 and the next inequality. Perhaps a better way to write the inequality, while keeping it short, would be to include an part in-between, like [average across G] $\ge$ [average across all means] $\ge$ RHS. But this is a minor detail, and I believe the proof is correct (small correction, line 162: should the referenced lemma be 2.3?).

A bit larger concerns are related to experemental evaluation and method significance. One strength is the amount of different methods/baselines. However:
* A sample size $m$ is introduced. But how does it relate to $a$, $b$, $\delta$ or $\varepsilon$? I believe I can see that in the experiment code, but the description is missing from the paper.
* Three fixed datasets are tested, where it is also shown that the empirical mean outperforms most methods. I strongly believe that testing on synthetic data sampled from more clean distributions would have been necessary for any conclusions, e.g. a large ground Gaussian/uniform sample. Also, not sure if accuracy should be plotted in log scale, or whether the accuracy valus in large sample sizes are not meaningfully distinct.
* Runtime barely grows with the sample size. Unless I'm misunderstanding what sample size is, should not it be essentially linear for introduced methods? Could the issue lie in vectorized numpy implementations with bottlenecks around the calls? Wouldn't it make sense to try an aforementioned synthetic dataset with a large sample size? And if it is not supposed to be linear, then again a clarification about sample size is important.

Last small question: line 179: what kind of non-optimality is spoken about here?

---

> ### Author Rebuttal · Authors · 2025-07-30
>
> We would like to thank the reviewer for the valuable comments and the raised questions. Due to NeurIPS 2025 restrictions, we’re unable to share links or images in the rebuttal or repository. We’ve included equivalent information in tabular form and kindly appreciate your understanding of these constraints.
>
> **Comment.** A sample size $m$ is introduced ... missing from the paper.
>
> **Response.** The sample size $m$ is set to $ab \varepsilon^{-1} \log \delta^{-1}$. To clarify, this is the result of having $b\log \delta^{-1}$ subsamples of size $a \varepsilon^{-1}$ each. We thank the reviewer for the comment and will add it to the paper explicitly.
>
> **Comment and Question.** Three fixed datasets ... Gaussian/uniform sample; While this is ... much larger sample sizes.
>
> **Response.** Please refer to the last response to reviewer otjC and tables therein.
>
> **Comment.** Also, not sure if accuracy should be plotted in log scale, or whether the accuracy values in large sample sizes are not meaningfully distinct.
>
> **Response.** Following the reviewer's suggestion, we plotted the accuracy in linear scale for both MNIST and the randomly generated dataset. Indeed, the accuracy values for large sample sizes essentially coincide (Tables 1 and 7).
>
> **Comment.** Runtime barely grows ... sample size is important.
>
> **Response.** We thank the reviewer for pointing this out. The runtime plots in the paper are shown in log-scale, which can visually compress differences and obscure linear growth, especially when the slope is small. When replotted in linear scale, the trends are indeed mildly increasing and consistent with near-linear growth, albeit with a small slope. We also repeated the experiments on the randomly generated synthetic dataset referenced above, and in that case the runtime growth is visibly linear. While we cannot upload plots, we have provided tables summarizing the above information (Tables 1 and 2 below, and 3 and 4 in reviewer otjC's last response). We believe the soft growth observed on the original dataset is largely due to implementation-level factors. In particular, our use of numpy's vectorized operations can lead to non-obvious runtime behavior, as such operations benefit from low-level optimizations (e.g., memory locality, multi-threaded backends) and often incur fixed overheads. As a result, runtime may appear sublinear over moderate input sizes, especially when memory access. Lastly, we agree that applying the methods to significantly larger datasets (as we did in the synthetic experiments) helps confirm the expected linear scaling more clearly, and we are happy to include these linear-scale plots and additional clarifications in the final version of the paper.
>
> **Table 1: MNIST — Accuracies vs. Sample Size**
>
> | $m$   | EM       | EM $\sigma$     | FGD      | FGD $\sigma$    | CWM      | CWM $\sigma$    | MSS      | MSS $\sigma$    | CSS      | CSS $\sigma$    | WT       | WT $\sigma$     |
> | ----- | -------- | -------- | -------- | -------- | -------- | -------- | -------- | -------- | -------- | -------- | -------- | -------- |
> | 10    | 1.098528 | 0.028635 | 1.155126 | 0.034458 | 1.175392 | 0.037500 | 1.392344 | 0.063439 | 1.001193 | 0.000251 | 1.050220 | 0.109802 |
> | 15    | 1.089786 | 0.022853 | 1.091481 | 0.021414 | 1.089386 | 0.019651 | 1.175579 | 0.041341 | 1.001193 | 0.000290 | 1.183068 | 0.040979 |
> | 20    | 1.050944 | 0.017004 | 1.071097 | 0.016169 | 1.079405 | 0.015386 | 1.150386 | 0.033795 | 1.001149 | 0.000299 | 1.166793 | 0.038011 |
> | 25    | 1.042582 | 0.012022 | 1.058092 | 0.013737 | 1.056672 | 0.011956 | 1.108955 | 0.023705 | 1.001170 | 0.000305 | 1.120348 | 0.024574 |
> | 30    | 1.035054 | 0.009981 | 1.042930 | 0.009194 | 1.045296 | 0.009135 | 1.091521 | 0.016239 | 1.001194 | 0.000290 | 1.097396 | 0.022739 |
> | 100   | 1.009560 | 0.002123 | 1.011862 | 0.002859 | 1.012232 | 0.002303 | 1.041629 | 0.011196 | 1.001229 | 0.000242 | 1.042088 | 0.011341 |
> | 200   | 1.004750 | 0.000965 | 1.007189 | 0.001480 | 1.007697 | 0.001541 | 1.022038 | 0.005945 | 1.001251 | 0.000228 | 1.025499 | 0.005615 |
> | 500   | 1.001954 | 0.000498 | 1.002961 | 0.000648 | 1.002704 | 0.000572 | 1.010661 | 0.002279 | 1.001248 | 0.000221 | 1.012654 | 0.002951 |
> | 1000  | 1.000921 | 0.000239 | 1.001301 | 0.000301 | 1.001286 | 0.000261 | 1.005201 | 0.001392 | 1.001310 | 0.000216 | 1.006786 | 0.001536 |
> | 2000  | 1.000493 | 0.000131 | 1.000700 | 0.000114 | 1.000717 | 0.000105 | 1.002898 | 0.000617 | 1.001290 | 0.000185 | 1.003791 | 0.000942 |
> | 5000  | 1.000180 | 0.000054 | 1.000294 | 0.000072 | 1.000270 | 0.000071 | 1.001383 | 0.000327 | 1.001360 | 0.000165 | 1.001820 | 0.000490 |
> | 10000 | 1.000089 | 0.000027 | 1.000142 | 0.000027 | 1.000151 | 0.000031 | 1.000692 | 0.000117 | 1.001340 | 0.000169 | 1.000996 | 0.000299 |
>
> **Table 2: MNIST — Runtimes vs. Sample Size**
>
> | $m$   | FGD      | FGD $\sigma$    | CWM      | CWM $\sigma$    | MSS      | MSS $\sigma$    | CSS      | CSS $\sigma$    | WT       | WT $\sigma$     |
> | ----- | -------- | -------- | -------- | -------- | -------- | -------- | -------- | -------- | -------- | -------- |
> | 10    | 0.331120 | 0.025323 | 0.303907 | 0.021514 | 0.305031 | 0.022829 | 0.752144 | 0.047994 | 0.498704 | 0.101033 |
> | 15    | 0.328824 | 0.029853 | 0.305248 | 0.023860 | 0.304551 | 0.019328 | 0.750669 | 0.052915 | 0.486801 | 0.095939 |
> | 20    | 0.321185 | 0.022370 | 0.313300 | 0.026494 | 0.304787 | 0.021680 | 0.753925 | 0.049201 | 0.469229 | 0.060214 |
> | 25    | 0.320447 | 0.024325 | 0.305469 | 0.018924 | 0.312867 | 0.026952 | 0.758950 | 0.052323 | 0.454810 | 0.045884 |
> | 30    | 0.324699 | 0.023393 | 0.306246 | 0.021164 | 0.307380 | 0.021195 | 0.754369 | 0.050649 | 0.484448 | 0.061920 |
> | 100   | 0.332200 | 0.029719 | 0.305576 | 0.022594 | 0.304266 | 0.021536 | 0.753291 | 0.047923 | 0.471533 | 0.064969 |
> | 200   | 0.339868 | 0.032710 | 0.307593 | 0.021598 | 0.306124 | 0.020822 | 0.753950 | 0.052971 | 0.472828 | 0.057820 |
> | 500   | 0.341225 | 0.032963 | 0.311128 | 0.020642 | 0.309915 | 0.021704 | 0.757828 | 0.039416 | 0.476949 | 0.061728 |
> | 1000  | 0.340164 | 0.029454 | 0.313098 | 0.025072 | 0.308668 | 0.021177 | 0.757372 | 0.050639 | 0.478237 | 0.062958 |
> | 2000  | 0.347602 | 0.029652 | 0.320347 | 0.026740 | 0.317822 | 0.022311 | 0.760508 | 0.050913 | 0.498500 | 0.058959 |
> | 5000  | 0.360183 | 0.035123 | 0.320959 | 0.022485 | 0.322097 | 0.022088 | 0.754888 | 0.051201 | 0.496606 | 0.064727 |
> | 10000 | 0.369807 | 0.032692 | 0.336506 | 0.023159 | 0.345213 | 0.029987 | 0.759944 | 0.047348 | 0.526814 | 0.062030 |
>
> **Question.** Last small question: line 179: what kind of non-optimality is spoken about here?
>
> **Response.** The coordinate-wise median is not optimal with respect to the convergence rate. The rate is of the form $\sqrt{\frac{Tr(\Sigma) \log \delta^{-1}}{n}}$ instead of the optimal $\sqrt{\frac{Tr(\Sigma)}{n}} + \sqrt{\frac{\lambda_{\max} \log \delta^{-1}}{n}}$. There are other algorithms which achieve optimality in the convergence rate like the \emph{median-of-means} tournaments. See the Lugosi and Mendelson survey for a detailed explanation.
>
> **Question.** I believe the clarity of Lemma 2.3 ... infinitesimally close to it.
>
> **Response.** Algorithm 2 does not always converge to the geometric median of the points. Take, for example, the triangle with coordinates $(0,1),  (0,-1), (1,0)$. The geometric median is the point $(1/\sqrt 3,0)$. If we initialize the algorithm at the origin, the algorithm never makes any progress and stays at the origin. We wish to reiterate that our goal was only to be sufficiently close to the geometric median. As mentioned in Remark 4.4, the state of the art geometric median algorithms have a significantly higher running time than what we are aiming for. Since even the best possible algorithm converging to the geometric median will run in time $\Omega(nd \log \varepsilon^{-1})$, a running time not achievable with current algorithms and a running time that is already superlinear in our setting, we can only hope to be sufficiently close.

---

> > ### Comment · Reviewer_1w8V · 2025-08-06
> >
> > I thank the authors for their detailed response to the questions and discussion. In my opinion, the additional experiments (presented in the reply to Reviewer otjC) and the clarifications to parts of the paper (including the figures and some definitions) will notably improve its quality. I have adjusted my grading accordingly.

---

### Official Review · Reviewer_otjC · 2025-07-03

**Clarity:** 4
**Significance:** 3
**Originality:** 3
**Rating:** 5
**Confidence:** 4

**Summary:**

The paper suggests techniques to compute an approximation to the mean of a given n points in an Euclidean space. The approximation is (1+epsilon) multiplicative factor, using only O(log(1/delta)/epsilon) i.i.d samples. The state of the art is  (1/(epsilon*delta)) is tight, but assuming that the algorithm is just "compute the mean of the sample". Instead, the authors suggest aggregation techniques and median approaches. Experiments on some datasets is also provided, as well as code.

**Questions:**

The idea of using the median of small samples for computing the global mean seems very related to the famous paper:
https://www.tau.ac.il/~nogaa/PDFS/amsz4.pdf

Please provide some comparison, at least for the algorithm itself.

**Ethical Concerns:**

["NO or VERY MINOR ethics concerns only"]

**Quality:**

3

**Strengths And Weaknesses:**

Strong:
-A fundamental problem in learning and statistics.
- Open code.
- Provable results.

Weak:
- The algorithm looks like a random version of a paper with a very similar goal that was appeared recently in NeuRIPS:
https://arxiv.org/abs/1906.04705
Although there are not so many papers regarding this problem, the paper is not cited and it seems that the authors are unaware of the result which essentially computes the mean by multiple aggregations. I am afraid that this fact alone suffices to reject this nice paper.

- The experiments are very poor. Why MNIST which is considered small and a "toy dataset"? Why not datasets of billions of points if the running time is sub-linear? Even random numbers would do..

---

> ### Author Rebuttal · Authors · 2025-07-30
>
> We would like to thank the reviewer for the useful comments, questions and suggestions. Moreover, we would like to highlight that, in accordance with the NeurIPS 2025 policy, we are not allowed to include any links in our rebuttal response, nor are we permitted to upload images—either in the rebuttal or to the already submitted repository. We are thus providing tables containing the same information as plots. We regret that you are unable to view the updated plots, and we kindly ask for your understanding and trust given these constraints.
>
> **Question.** The idea ... the algorithm itself.
>
> **Response.** The idea of median-of-means estimator is indeed old. Well-known references include (Nemirovski and Yudin, 1983; Jerrum et al., 1986 including the paper the reviewer has cited - Alon et al., 1996). Our work focuses on adapting the idea for higher dimensions. There are indeed many options for this including coordinate-wise median, geometric median among others. The proofs of convergence and running time we provide are novel to the best of our knowledge. Specifically, the best known algorithm for computing (approximate) geometric median (Cohen et al. 2016) leads to a much worse running time of the algorithm.
>
> **Comment.** The algorithm looks like ... reject this nice paper.
>
> **Response.** The paper in reference, though dealing with problems of a  similar flavor, has no direct connection to the problem we aim to solve. The paper aims at computing loss-less summaries for the mean. Sublinear algorithms necessarily require some form of approximation to the mean. The difference is also highlighted in the running time: their algorithm has complexity $O(nd + d^4 \log n)$. It is unlikely it could be used at any stage of the mean aggregation. But even if it could be used, its running time would be significantly worse that what we are aiming for.  However, we will add a reference and explain the differences in the final version.
>
> **Comment.** The experiments are very poor ... random numbers would do.
>
> **Response.** Following the Reviewers otjC and 1w8V’s suggestions, we ran additional experiments with larger sample sizes. Specifically, we generated a synthetic dataset from a 200-dimensional multivariate Gaussian distribution and extended the sample size up to $5 \times 10^5$.
>
> The results are in line with theory: They show that the error decreases sharply as the sample size grows and that the runtime grows linearly with sample size (Tables 3 and 4).
>
> Unfortunately, we were unable to go beyond this sample size due to memory limitations in our environment, where the process exhausts the available RAM. We will add a note in the final version explaining this limitation. We hope that the extended results are sufficient to address the reviewers’ concerns.
> We have also generated two plots illustrating how accuracy and runtime vary with dimensionality: We elaborated more on this aspect in the response to Reviewer sRHN.
> We will add these new experimental results to the final version of the paper.
>
>
> **Table 3: Synthetic Dataset ($d = 200$) — Accuracies vs. Sample Size**
>
> |    $m$ |     EM |            EM $\sigma$ |    FGD |          FGD $\sigma$ |    CWM |          CWM $\sigma$ |    MSS |          MSS $\sigma$ |    CSS |           CSS $\sigma$ |     WT |           WT $\sigma$ |
> | -----: | -----: | ---------------------: | -----: | --------------------: | -----: | --------------------: | -----: | --------------------: | -----: | ---------------------: | -----: | --------------------: |
> |     10 | 1.0000 |  $1.17 \times 10^{-1}$ | 1.1274 | $3.42 \times 10^{-2}$ | 1.1276 | $3.49 \times 10^{-2}$ | 1.2528 | $2.42 \times 10^{-2}$ | 1.0000 | $1.11 \times 10^{-16}$ | 1.2587 | $2.19 \times 10^{-2}$ |
> |     15 | 1.0000 |  $8.88 \times 10^{-2}$ | 1.0620 | $1.22 \times 10^{-2}$ | 1.0671 | $1.31 \times 10^{-2}$ | 1.1427 | $1.29 \times 10^{-2}$ | 1.0000 |                 $0.00$ | 1.1442 | $1.34 \times 10^{-2}$ |
> |     20 | 1.0000 |  $9.42 \times 10^{-2}$ | 1.0549 | $1.05 \times 10^{-2}$ | 1.0589 | $9.72 \times 10^{-3}$ | 1.1195 | $1.29 \times 10^{-2}$ | 1.0000 |                 $0.00$ | 1.1250 | $1.17 \times 10^{-2}$ |
> |     25 | 1.0000 |  $6.08 \times 10^{-2}$ | 1.0399 | $5.84 \times 10^{-3}$ | 1.0393 | $7.14 \times 10^{-3}$ | 1.0873 | $8.39 \times 10^{-3}$ | 1.0019 |  $8.89 \times 10^{-4}$ | 1.0915 | $8.92 \times 10^{-3}$ |
> |     30 | 1.0000 |  $2.72 \times 10^{-2}$ | 1.0316 | $5.53 \times 10^{-3}$ | 1.0299 | $4.48 \times 10^{-3}$ | 1.0672 | $6.72 \times 10^{-3}$ | 1.0013 |  $9.67 \times 10^{-5}$ | 1.0708 | $5.76 \times 10^{-3}$ |
> |    100 | 1.0000 |  $6.47 \times 10^{-2}$ | 1.0088 | $1.32 \times 10^{-3}$ | 1.0091 | $1.18 \times 10^{-3}$ | 1.0302 | $2.62 \times 10^{-3}$ | 1.0005 |  $3.69 \times 10^{-5}$ | 1.0313 | $2.94 \times 10^{-3}$ |
> |    200 | 1.0000 |  $3.85 \times 10^{-2}$ | 1.0058 | $7.01 \times 10^{-4}$ | 1.0057 | $6.43 \times 10^{-4}$ | 1.0194 | $2.18 \times 10^{-3}$ | 1.0003 |  $2.54 \times 10^{-5}$ | 1.0202 | $1.97 \times 10^{-3}$ |
> |    500 | 1.0000 |  $1.66 \times 10^{-2}$ | 1.0022 | $2.49 \times 10^{-4}$ | 1.0022 | $2.41 \times 10^{-4}$ | 1.0095 | $7.99 \times 10^{-4}$ | 1.0001 |  $1.03 \times 10^{-5}$ | 1.0102 | $8.94 \times 10^{-4}$ |
> |   1000 | 1.0000 |                 $0.00$ | 1.0011 | $9.79 \times 10^{-5}$ | 1.0011 | $9.97 \times 10^{-5}$ | 1.0048 | $4.04 \times 10^{-4}$ | 1.0001 |  $6.40 \times 10^{-6}$ | 1.0051 | $4.65 \times 10^{-4}$ |
> |   2000 | 1.0000 |  $5.44 \times 10^{-2}$ | 1.0006 | $6.73 \times 10^{-5}$ | 1.0006 | $7.67 \times 10^{-5}$ | 1.0028 | $2.76 \times 10^{-4}$ | 1.0001 |  $3.81 \times 10^{-6}$ | 1.0031 | $2.75 \times 10^{-4}$ |
> |   5000 | 1.0000 |  $9.93 \times 10^{-3}$ | 1.0002 | $2.73 \times 10^{-5}$ | 1.0002 | $2.63 \times 10^{-5}$ | 1.0013 | $1.07 \times 10^{-4}$ | 1.0001 |  $2.04 \times 10^{-6}$ | 1.0014 | $1.47 \times 10^{-4}$ |
> |  10000 | 1.0000 | $1.17 \times 10^{-10}$ | 1.0001 | $1.21 \times 10^{-5}$ | 1.0001 | $1.04 \times 10^{-5}$ | 1.0007 | $6.10 \times 10^{-5}$ | 1.0001 |  $1.59 \times 10^{-6}$ | 1.0008 | $7.20 \times 10^{-5}$ |
> | 100000 | 1.0000 | $3.14 \times 10^{-18}$ | 1.0000 | $1.43 \times 10^{-6}$ | 1.0000 | $1.27 \times 10^{-6}$ | 1.0001 | $6.64 \times 10^{-6}$ | 1.0000 |  $6.10 \times 10^{-7}$ | 1.0001 | $1.00 \times 10^{-5}$ |
> | 500000 | 1.0000 | $3.14 \times 10^{-18}$ | 1.0000 | $2.92 \times 10^{-7}$ | 1.0000 | $2.52 \times 10^{-7}$ | 1.0000 | $2.00 \times 10^{-6}$ | 1.0000 |  $3.98 \times 10^{-7}$ | 1.0000 | $2.00 \times 10^{-6}$ |
>
> **Table 4: Synthetic Dataset ($d = 200$) — Runtimes vs. Sample Size**
>
> |    $m$ |      FGD |          FGD $\sigma$ |      CWM |          CWM $\sigma$ |      MSS |          MSS $\sigma$ |      CSS |          CSS $\sigma$ |       WT |           WT $\sigma$ |
> | -----: | -------: | --------------------: | -------: | --------------------: | -------: | --------------------: | -------: | --------------------: | -------: | --------------------: |
> |     10 | 0.023158 | $2.01 \times 10^{-3}$ | 0.000424 | $1.98 \times 10^{-4}$ | 0.000495 | $6.10 \times 10^{-5}$ | 0.000229 | $1.10 \times 10^{-5}$ | 0.000466 | $2.60 \times 10^{-5}$ |
> |     15 | 0.023287 | $2.97 \times 10^{-3}$ | 0.000434 | $3.57 \times 10^{-4}$ | 0.000663 | $3.94 \times 10^{-4}$ | 0.000339 | $2.44 \times 10^{-4}$ | 0.000701 | $5.38 \times 10^{-4}$ |
> |     20 | 0.016261 | $4.85 \times 10^{-3}$ | 0.000245 | $5.00 \times 10^{-5}$ | 0.000349 | $2.80 \times 10^{-5}$ | 0.000164 | $1.80 \times 10^{-5}$ | 0.000338 | $3.30 \times 10^{-5}$ |
> |     25 | 0.014721 | $2.97 \times 10^{-3}$ | 0.000273 | $6.80 \times 10^{-5}$ | 0.000385 | $7.80 \times 10^{-5}$ | 0.000167 | $8.00 \times 10^{-6}$ | 0.000392 | $9.60 \times 10^{-5}$ |
> |     30 | 0.015118 | $3.30 \times 10^{-3}$ | 0.000248 | $2.50 \times 10^{-5}$ | 0.000353 | $2.10 \times 10^{-5}$ | 0.000173 | $1.20 \times 10^{-5}$ | 0.000364 | $3.60 \times 10^{-5}$ |
> |    100 | 0.017072 | $3.41 \times 10^{-3}$ | 0.000379 | $6.40 \times 10^{-5}$ | 0.000439 | $1.90 \times 10^{-5}$ | 0.000282 | $4.30 \times 10^{-5}$ | 0.000625 | $2.20 \times 10^{-5}$ |
> |    200 | 0.019220 | $3.32 \times 10^{-3}$ | 0.000522 | $6.40 \times 10^{-5}$ | 0.000655 | $4.50 \times 10^{-5}$ | 0.000479 | $5.80 \times 10^{-5}$ | 0.001034 | $9.30 \times 10^{-5}$ |
> |    500 | 0.021631 | $3.98 \times 10^{-3}$ | 0.000835 | $5.20 \times 10^{-5}$ | 0.001078 | $1.04 \times 10^{-4}$ | 0.000850 | $4.90 \times 10^{-5}$ | 0.002033 | $1.54 \times 10^{-4}$ |
> |   1000 | 0.022246 | $2.99 \times 10^{-3}$ | 0.001404 | $7.10 \times 10^{-5}$ | 0.001841 | $4.40 \times 10^{-4}$ | 0.001629 | $2.26 \times 10^{-4}$ | 0.003663 | $3.48 \times 10^{-4}$ |
> |   2000 | 0.025202 | $3.42 \times 10^{-3}$ | 0.002930 | $5.24 \times 10^{-4}$ | 0.003299 | $3.50 \times 10^{-4}$ | 0.002832 | $2.04 \times 10^{-4}$ | 0.007801 | $1.19 \times 10^{-3}$ |
> |   5000 | 0.047089 | $6.56 \times 10^{-3}$ | 0.010890 | $9.59 \times 10^{-4}$ | 0.008105 | $5.00 \times 10^{-4}$ | 0.010210 | $1.04 \times 10^{-3}$ | 0.018438 | $1.52 \times 10^{-3}$ |
> |  10000 | 0.042100 | $3.96 \times 10^{-3}$ | 0.016434 | $9.72 \times 10^{-4}$ | 0.016744 | $1.03 \times 10^{-3}$ | 0.022541 | $7.60 \times 10^{-4}$ | 0.037705 | $2.83 \times 10^{-3}$ |
> | 100000 | 0.256733 | $2.73 \times 10^{-2}$ | 0.227310 | $2.16 \times 10^{-2}$ | 0.223843 | $1.91 \times 10^{-2}$ | 0.339804 | $2.20 \times 10^{-2}$ | 0.483566 | $7.13 \times 10^{-2}$ |
> | 500000 | 1.321969 | $1.16 \times 10^{-1}$ | 1.278243 | $1.11 \times 10^{-1}$ | 1.277424 | $1.12 \times 10^{-1}$ | 1.695665 | $1.05 \times 10^{-1}$ | 2.782799 | $3.10 \times 10^{-1}$ |

---

> > ### Comment · Reviewer_otjC · 2025-08-07
> >
> > I thank the authors for adding the experimental results, and would encourage the other reviewers to accept it.
> > Meanwhile, I noticed that there are many related results that are not cited, including papers by Jeff M. Phillips (e.g.,
> > "Near-Optimal Coresets of Kernel Density Estimate"). See many more in a survey on this problem (that is also not cited):
> > @article{DBLP:journals/corr/abs-2111-03046,
> >   author       = {Alaa Maalouf and
> >                   Ibrahim Jubran and
> >                   Dan Feldman},
> >   title        = {Introduction to Coresets: Approximated Mean},
> >   journal      = {CoRR},
> >   volume       = {abs/2111.03046},
> >   year         = {2021},
> >   eprinttype    = {arXiv},
> >   bibsource    = {dblp computer science bibliography, https://dblp.org}
> > }

---

> > > ### Author Response · Authors · 2025-08-08
> > >
> > > We thank the reviewer for the references and the insightful feedback, in particular the pointer to the approximate coresets for the mean which is indeed very relevant. We will incorporate them into the final version of the paper, adding a detailed comparison between them and our work.

---

### Note · Authors · 2025-08-13

We thank the reviewers for their responses and for a productive discussion. We would like to summarize and underscore a few things from the discussion as part of our final remarks.

First, we would like to highlight that our work gives us a sublinear algorithm for mean estimation with optimal sample complexity and optimal running time. Our work improves over the previous works by $\log$ factors in both $1/\varepsilon$ and $1/\delta$. Note that in the area of Learning theory, improving by log factors is often non-trivial, for example the literature spent several decades and nearly a dozen papers removing one $\log \varepsilon^{-1}$ to achieve optimal PAC learning bounds.

Second, we thank the reviewers for relevant additional citations.

Finally, we thank the reviewers for their suggestions, which led to a series of additional experiments. Our results indicate that the error sharply decreases as the sample size grows, and that the runtime scales linearly with sample size—a finding we confirmed on a large synthetic dataset. Regarding dimensionality, the accuracies of our estimators (coordinate-wise median and fast gradient descent) remained low and stable, while the runtime exhibited a roughly linear growth with dimension, consistent with theory.

---

### Decision · Program_Chairs · 2025-09-17

**Decision:**

Accept (poster)

**Comment:**

This paper studies the multivariate mean estimation problem. It shows an aggregate estimator that averages mean estimates on bootstrap samples (with independent and uniform sampling) can output an $(1+\varepsilon)$-approximate mean with probability at least $1 - \delta$ by using  $O(\varepsilon^{-1} \log \delta^{-1})$ sampled points. This is an improvement of the classic  $O(\varepsilon^{-1} \delta^{-1})$ sample complexity and the SOTA $O(\varepsilon^{-1} \log \delta^{-1} (\log \varepsilon^{-1} + \log \delta^{-1}))$ sample complexity. The paper further presents three efficient algorithms that run the aggregate estimator in $O(\varepsilon^{-1} \log \delta^{-1})$ time. Experimental results on a real-world data set shows one of the proposed estimators achieve the best accuracy-runtime tradeoff, while the other two have competitive performance compared to the SOTA method.

---

The paper makes a plausible improvement on SOTA sample complexity for mean estimation, and provides efficient estimators.

What holds reviewers Vydg and sRHN back is the limited technical novelty. Reviewer Vydg also points out the improvement only applies to the $\ell_2$ norm while prior results apply to $\ell_p$. Another notable weakness is the experiment section. Concerns include the limited scale of the data set (reviewers otjC & 1w8V & sRHN), insufficient comparison with empirical mean as a baseline (reviewer sRHN) and missing verification of the dimension-free property (reviewer sRHN).

Despite those concerns, most reviewers are on the positive side. I think the work is decent and the paper is well written. Its contribution may have a relatively narrowed scope, but its merits outweigh this limitation. Authors should incorporate all reviewers' feedback in revision.